# Glycolysis regulates KRAS plasma membrane localization and function through defined glycosphingolipids

Junchen Liu[1], Ransome van der Hoeven[2], Walaa E. Kattan [1], Jeffrey T. Chang [1,3], Dina Montufar-Solis[1], Wei Chen[1], Maurice Wong [4], Yong Zhou [1,3], Carlito B. Lebrilla [4] & John F. Hancock [1,3] ✉

Oncogenic KRAS expression generates a metabolic dependency on aerobic glycolysis, known as the Warburg effect. We report an effect of increased glycolytic flux that feeds into glycosphingolipid biosynthesis and is directly linked to KRAS oncogenic function. High resolution imaging and genetic approaches show that a defined subset of outer leaflet glycosphingolipids, including GM3 and SM4, is required to maintain KRAS plasma membrane localization, with GM3 engaging in cross-bilayer coupling to maintain inner leaflet phosphatidylserine content. Thus, glycolysis is critical for KRAS plasma membrane localization and nanoscale spatial organization. Reciprocally oncogenic KRAS selectively upregulates cellular content of these same glycosphingolipids, whose depletion in turn abrogates KRAS oncogenesis in pancreatic cancer models. Our findings expand the role of the Warburg effect beyond ATP generation and biomass building to high-level regulation of KRAS function. The positive feedforward loop between oncogenic KRAS signaling and glycosphingolipid synthesis represents a vulnerability with therapeutic potential.

NRAS, HRAS, and KRAS, are small GTPases that cycle between inactive GDP-bound and active GTP-bound states to regulate cell growth. Point mutations that lock KRAS into the GTP-bound state occur with high frequency in pancreatic, colon, and non-small lung cancers. This prominence of KRAS as an important oncogenic driver renders urgent the need to decode its biological and biophysical vulnerabilities. For biological activity, KRAS must be both localized to the plasma membrane (PM) and spatially organized into protein-lipid assemblies known as nanoclusters[1–5]. KRAS PM interactions are mediated by a bi-partite anchor consisting of a C-terminal S-farnesyl cysteine carboxyl-methylester and an adjacent polybasic domain of six contiguous lysines[6,7]. This KRAS C-terminal anchor encodes exquisite lipid binding specificity for mixed chain

phosphatidylserine (PtdSer) species[8,9], which therefore renders KRAS PM interactions absolutely dependent on phosphatidylserine content and distribution[9–14].

Oncogenic KRAS enhances aerobic glycolysis by increasing expression and activity of glucose transporters and glycolytic enzymes including hexokinases[15–18]. Metabolic intermediates of aerobic glycolysis in turn supply anabolic processes generating proteins, lipids, and nucleotides to support rapid cell proliferation[19]. An early derivative of glucose-6-P, the first product of glycolysis, is UDP-glucose (Fig. 1a), which is used for protein and lipid glycosylation, including the synthesis of glycosphingolipids (GSLs). GSLs are resident on the outer leaflet of the PM and exhibit dysregulated expression in many cancers[20–23]. Given evidence that

[1]Department of Integrative Biology and Pharmacology, McGovern Medical School, University of Texas Health Science Center, Houston, TX, USA. [2]Department of Diagnostic and Biomedical Sciences, School of Dentistry, The University of Texas Health Science Center at Houston, Houston, TX, USA. [3]Graduate School of Biological Sciences, M. D. Anderson Cancer Center and University of Texas Health Science Center, Houston, TX, USA. [4]Department of Chemistry, University of California, Davis, CA, USA. ✉e-mail: john.f.hancock@uth.tmc.edu

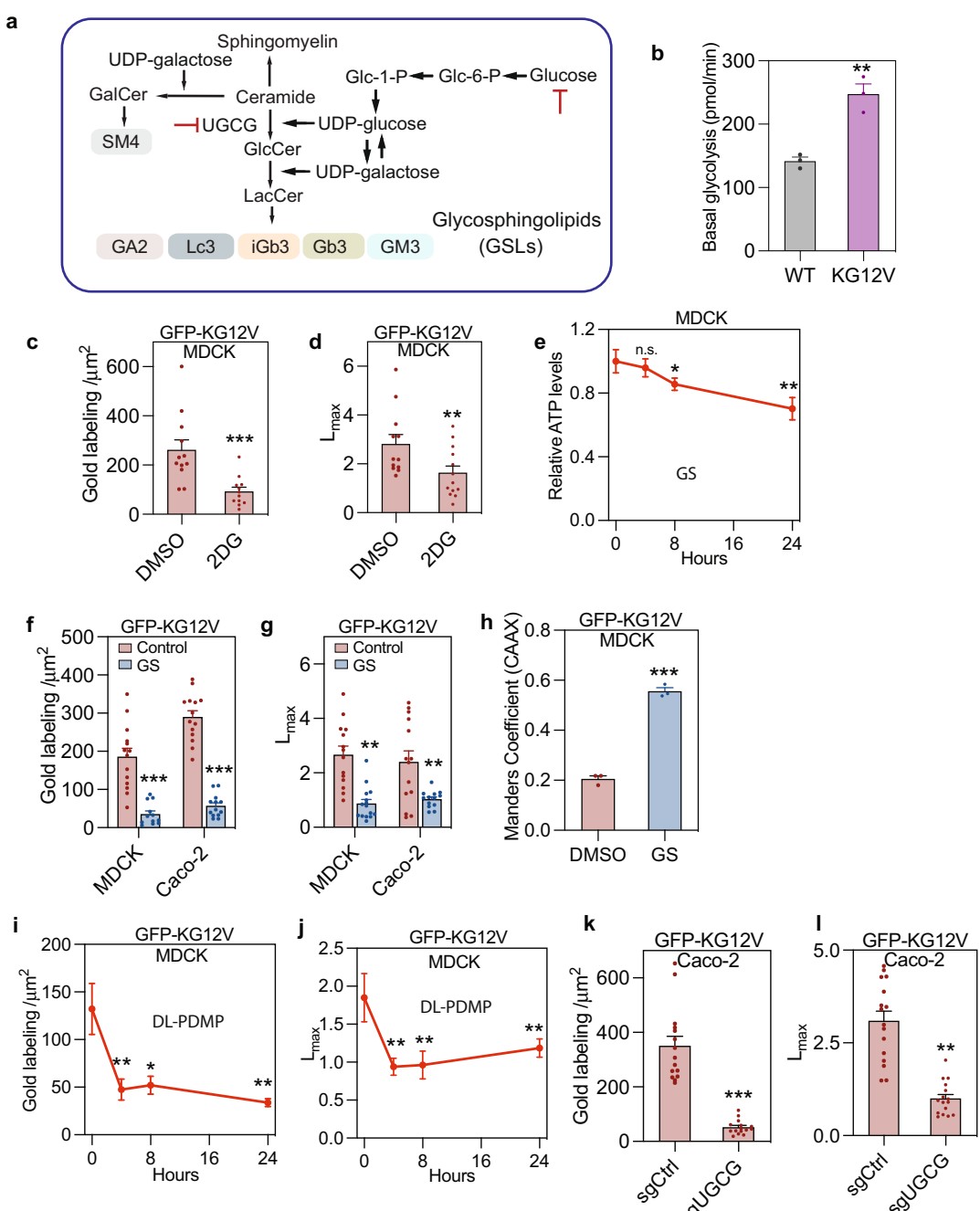

**Fig. 1 | Suppressing glycolysis mislocalizes KRASG12V from the plasma membrane.** **a** Diagram of GSL synthetic pathways. UGCG, UDP-glucose ceramide glucosyltransferase. **b** Basal glycolysis rate of WT and KRASG12V (KG12V) expressing MDCK cells measured in a Seahorse XF analyzer (±SEM, $n = 3$ independent biological relicates, **$p < 0.01$). Time course of changes in 2DG (10 mM)-treated MDCK cells of: GFP-KRASG12V (KG12V) PM binding quantified by EM as mean gold labeling density (**c**), and nanoclustering quantified as $L_{max}$ (**d**) ($n = 12$ PM sheets for each group). For the DMSO control time course see Fig. S1J–L. Time course of changes in relative ATP levels for control ($t = 0$) and glucose-depleted (GS) MDCK-KRASG12V cells (±SD, $n = 3$ wells, *$p < 0.05$, **$p < 0.01$) (**e**). PM localization (**f**) and nanoclustering (**g**) of KRASG12V-expressing MDCK ($n = 14$ for Control, $n = 12$ for GS) or Caco-

2 cells, evaluated by EM, after 4 h of glucose depletion ($n = 14$ for Control, $n = 13$ for GS). Time course of GFP-KRASG12V mislocalization in glucose-depleted MDCK cells co-expressing mCherry-CAAX (an endomembrane marker) evaluated using Manders coefficients (±SD, $n = 3$ independent replicates, ***$p < 0.001$) (**h**). Time course of changes in PM binding (**i**) and nanoclustering (**j**) of GFP-KRASG12V in DL-PDMP-treated MDCK cells. EM evaluation of PM binding (**k**) and nanoclustering (**l**) of GFP-KRASG12V in UGCG knockdown (sgUGCG) and non-targeting control (sgCtrl) Caco-2 cells ($n = 41$ PM sheets). In (**c, d**), (**f, g**), and (**i–l**) data are mean ± SEM. Differences in gold labeling densities and $L_{max}$ were evaluated in two-tailed Student's t-tests and bootstrap tests respectively (*$p < 0.05$, **$p < 0.01$, ***$p < 0.001$). Exact $p$-values and source data are provided as a Source data file.

glucose availability regulates cellular GSL levels, and the dependence of KRAS membrane targeting on PM lipid content, we explored a possible reciprocal interdependence of KRAS function with anaerobic glycolysis in the context of GSL metabolism.

In this study, we combine high-resolution imaging and genetic approaches, to show that defined GSLs are essential for the maintenance of KRAS PM localization and KRAS oncogenicity. GSL metabolism is therefore a vulnerability of KRAS mutant cancers that may have therapeutic implications.

## Results

### Glycolysis is required for KRASG12V PM localization

Consistent with previous reports[24] expression of KRASG12V, a constitutively activated GTP-bound mutant of KRAS, in Madin-Darby Canine Kidney (MDCK) cells significantly increased the basal glycolytic rate and lactate production rate compared to cognate wild-type (WT) cells (Figs. 1b and S1a). To test whether glycolysis is required for KRAS PM localization, we treated MDCK cells stably expressing GFP-KRASG12V with 2-deoxyglucose (2DG), a glycolysis inhibitor. Intact PM sheets prepared from these cells were labeled with 4.5 nm gold conjugated anti-GFP antibodies, followed by EM imaging and spatial statistical analysis using univariate K-functions expressed as $L(r)-r$. The level of PM localized KRASG12V, quantified by the number of gold particles per μm², and the extent of KRASG12V nanoclustering, quantified by the maximal value ($L_{max}$) of the $L(r)-r$ function, were both significantly reduced after 4 h incubation with 2DG (Fig. 1c, d). Next, we subjected MDCK cells to 4 h of glucose deprivation. Cellular ATP levels were unaffected at this time point (Fig. 1e), however, EM analysis showed extensive loss of KRASG12V from the PM and decreased clustering of KRASG12V remaining on the PM (Fig. 1f, g). Identical results were observed in Caco-2 cells, a human colon cancer cell line (Fig. 1f, g). These results indicate that acute glucose deprivation displaced KRASG12V from the PM independent of depleting cellular ATP. Concordantly confocal microscopy of 4hr-glucose-starved MDCK cells co-expressing GFP-KRASG12V and mCherry-CAAX, a general endomembrane marker, showed a significant loss of KRAS from the PM, as quantified by Manders coefficients (Figs. 1h and S1b), which was rapidly and completely restored by glucose addback (Fig. S1c). The mechanism involves the KRASG12V anchor, not the G-domain, because GFP-appended with the minimal membrane anchor of KRAS (GFP-tK) exhibited identical behavior in glucose-depleted cells (Fig. S1d, e). Together, these results indicate that glycolysis is required for KRAS PM localization.

### Glycosphingolipid synthesis mediates glycolysis-regulated KRASG12V membrane targeting

To directly examine whether GSL biosynthesis is required for KRAS PM targeting, we first treated MDCK cells expressing GFP-KRASG12V with DL-PDMP, an inhibitor of UDP-glucosyl transferase (UGCG), the enzyme that glycosylates ceramide using UDP-Glucose (Fig. 1a). DL-PDMP treatment rapidly and significantly reduced both KRASG12V PM localization and nanoclustering (Fig. 1i, j). Similarly, UGCG gene deletion in Caco-2 cells using CRISPR-cas9 significantly reduced GFP-KRASG12V PM localization and nanoclustering (Fig. 1k, l). KRAS PM localization is critical for RAF PM recruitment and activation of the MAPK cascade, concordantly Caco-2 cells depleted of UGCG exhibited reduced levels of phosphorylated ERK (p-ERK) and MEK (p-MEK) (Fig. S1f). Next, we conducted lipid addback experiments under conditions of glucose starvation, DL-PDMP treatment, or UGCG knockout. Addback of glucosyl-ceramide (GlcCer) and lactosyl-ceramide (LacCer), but not galactosyl-ceramide (GalCer) recovered the PM localization and nanoclustering of KRASG12V in glucose starved MDCK cells (Fig. 2a, b). In DL-PDMP treated cells, LacCer, but neither GlcCer, or GalCer recovered the PM localization and nanoclustering of KRASG12V, likely reflecting the capacity of DL-PDMP to inhibit lactosyl-ceramide synthase (LCS) as well as UGCG[25] and therefore the requirement for the enzyme product of LCS to rescue GSL synthesis (Fig. 2a, b). This interpretation is supported by results in UGCG knockout Caco-2 cells, where GlcCer and LacCer, but not GalCer recovered the PM localization and nanoclustering of KRASG12V, mirroring results with glucose starvation (Fig. 2c, d). GalCer addback is an important control, because GalCer is not synthesized by UGCG and is not depleted by glucose starvation, or UGCG inhibition (Fig. 1a). Taken together, these data indicate that glycolysis controls KRASG12V localization and nanoclustering by regulating UGCG-dependent GSL synthesis.

Since PM PtdSer mediates KRAS PM interactions we examined whether GSLs might control PM PtdSer content. Acute glucose depletion or treatment with the UGCG inhibitor DL-PDMP markedly reduced PM localization of GFP-LactC2, a PtdSer probe by both EM and confocal analysis (Figs. 2e and S1g, h, i). This was unrelated to flippase activity because annexin V staining to measure outer leaflet PtdSer content did not increase with glucose starvation (Fig. S2a). The reduction of PtdSer on the PM was not due to changes in total PtdSer levels as revealed by the lipidomic analysis (Fig. S2b). Addback of GlcCer and LacCer, but not GalCer markedly raised the PM PtdSer content in glucose-depleted, or DL-PDMP-treated MDCK cells, as did addback of PtdSer (Fig. 2e). Finally, addback of PtdSer to glucose-starved, or DL-PDMP-treated MDCK cells (Fig. 2a), or glucose-starved, or UGCG knockout Caco-2 (Fig. 2c) cells fully restored KRASG12V to the PM, whereas addback of another phospholipid, PIP₂, had no effect. Nanoclustering of GFP-KRASG12V in both cells lines was also recovered by addback of PtdSer, but not PIP₂ (Fig. 2b, d). Collectively, these results indicate that GSL biosynthesis regulates PM PtdSer content, which is essential for KRAS PM interactions.

### KRASG12V expression enhances GSL synthesis

The data in Fig. 2 directly implicate GSL metabolism in KRAS PM binding. We next asked if KRAS reciprocally modulates the activity of GSL, or other, biosynthetic pathways relevant to membrane structure. We used shot-gun lipidomics to identify lipids whose levels differed consistently, and to the greatest fold extent, across multiple replicates of monoclonal MDCK cell lines differing only in expression of KRASG12V (Fig. 3a). Phosphatidylcholine (PC) levels increased significantly in KRASG12V cells as did total glycerophospholipid (GPL) levels (Fig. S3a, b), likely reflecting the role of PC as precursor for other classes of GPL. There was a decrease in PC and PE species with shorter acyl chains in favor of species with longer acyl chains (Fig. 3a) coupled with increased acyl chain saturation: GPLs with 4 to 6 double bonds decreased, whereas GPLs with 1 to 3 double bonds increased (Fig. S3c), upon KRASG12V expression. Overall PtdSer levels were not significantly different between each pair of matched cell lines, except for two PtdSer species (PS 18:1-20:0 and PS 18:0-20:1) that increased ~4-fold in the KRASG12V cells (Fig. 3a). These asymmetric PtdSer species are preferentially bound by the KRAS membrane anchor[8,26]. Verifying the generality of this KRASG12V reprogramming of the cellular lipidome, qualitatively similar, albeit quantitatively greater changes were observed in Caco-2 cells (Figs. S3a–d, and S4d), where unlike in MDCK cells, expression of KRASG12V enhanced the proliferation rate relative to the parental cell line (Fig. S3d). Most relevant to GSL metabolism, Ceramide (Cer) levels increased significantly upon KRASG12V expression in both MDCK and Caco-2 cells (Fig. 3a, b). There was also a switch from sphingolipids (SLs) with 34 carbons to SLs with 36 carbons and longer chain SLs with 42 and 44 carbons in both MDCK and Caco-2 cells expressing KRASG12V (Fig. S3e). MDCK and Caco-2 cells expressing KRASG12V in addition showed a significant reduction in Hexosyl-Ceramide (HexCer) levels (Fig. 3a, c), which include both GlcCer and GalCer as these species cannot be distinguished by mass-spectrometry (MS). Glucose starvation, or DL-PDMP treatment, further reduced HexCer levels (Fig. 3c). In contrast, glucose starvation did not change ceramide levels in KRASG12V-expressing cells (Fig. S4a), again indicating that conversion from ceramide to GlcCer is inhibited by glucose starvation. Since the steady state of this metabolic intermediate does not necessarily track with changes in flux, we measured the rate of generation of C¹³-labeled HexCer following isotopic labeling with C¹³ glucose (Fig. 3d, e). Carbons derived from C¹³-glucose, were rapidly accumulated into HexCer at 1-h and 4-h post labeling. KRASG12V expression in MDCK cells approximately doubled the rate of C¹³-labeling of HexCer at both time points (Fig. 3d). KRASG12V expression similarly significantly increased C¹³-labeling of HexCer in Caco-2 cells compared to parental cells (Fig. 3e); although the different magnitude

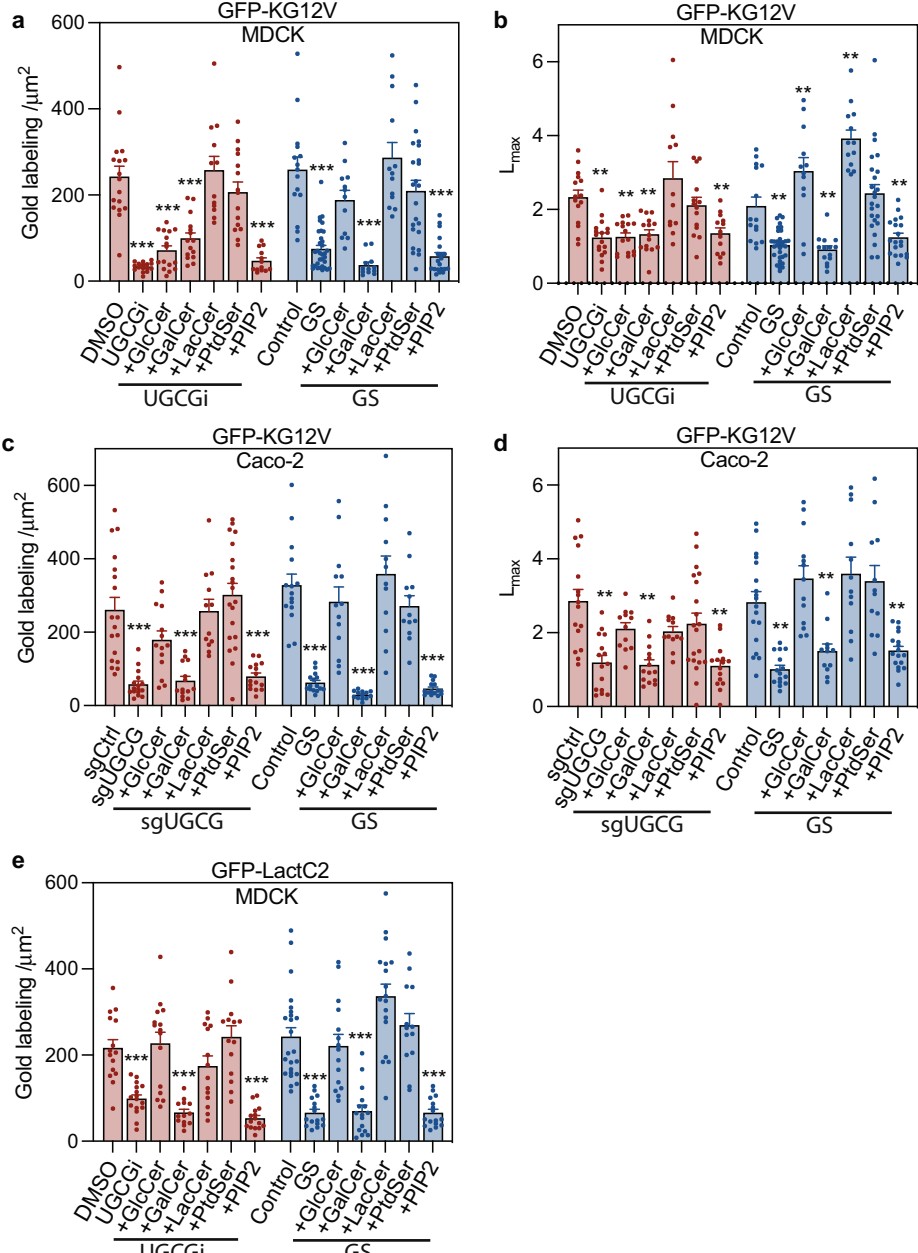

**Fig. 2 | GSLs are required for KRAS plasma membrane localization.**
**a**, **b** KRASG12V-expressing MDCK cells (KG12V) treated with 25 µM DL-PDMP (UGCGi) (24 h, (n = 14 PM sheets), or cultured in glucose-depleted (GS) medium (4 h, n = 16) and incubated with exogenous GalCer (n = 13), GlcCer (n = 15), LacCer (n = 14), PtdSer (n = 14) or PIP₂ (n = 14) for 1 h were evaluated using EM. KRASG12V PM binding was quantified as mean gold labeling density and nanoclustering as $L_{max}$. Comparisons are with cognate untreated control cells. **c**, **d** Identical GSL addback experiments and analyses to (**a**) and (**b**) in KRASG12V-expressing Caco-2. **e** Identical GSL addback experiments to (**a**) in GFP-LactC2-expressing MDCK cells. For (**a**–**e**) data are means ± SEM. Significance of differences between $L_{max}$ values for GSL-addback and control cells were evaluated in bootstrap tests (**p < 0.01) and differences in gold labeling density in two-tailed Student's t-tests (**p < 0.01, ***p < 0.001). Exact p-values and source data are provided as a Source data file.

of %C¹³-labeled HexCer indicates that the metabolic rate for this biosynthesis is different in the two cell lines; nevertheless, in both MDCK and Caco-2 cells expression of KRASG12V enhanced flux of glucose and ceramide into GSL biosynthesis.

Next we quantified the glycolipidomes of MDCK and Caco-2 cells using a modified MS procedure[27]. LacCer levels increased in KRASG12V-expressing cells relative to the cognate parental cells, suggesting that KRASG12V promotes the synthesis of downstream GSLs (Fig. 3f), consistent with the lipidomics and isotopic tracing data. GM3 also increased significantly in KRASG12V-expressing MDCK and Caco-2 cells (Fig. 3f). These upregulated GM3 species

have a similar acyl chain composition (42:1 and 42:2) to the most significantly changed HexCer species identified by isotopic tracing, indicating the biosynthetic pathway (Fig. S4b, c). SM4, a derivative of GalCer produced by UGT8, was also consistently upregulated, whereas Gb3 levels were reduced upon KRASG12V expression (Fig. 3f). Variable changes were observed in other GSLs (Fig. 3f) likely reflecting the intrinsic complexity and heterogeneity of GSLs in cells of different origins and growth states[27–30]. Taking these results together we conclude that KRASG12V-expressing cells prioritize routing of HexCer and LacCer into a specific cohort of GSLs.

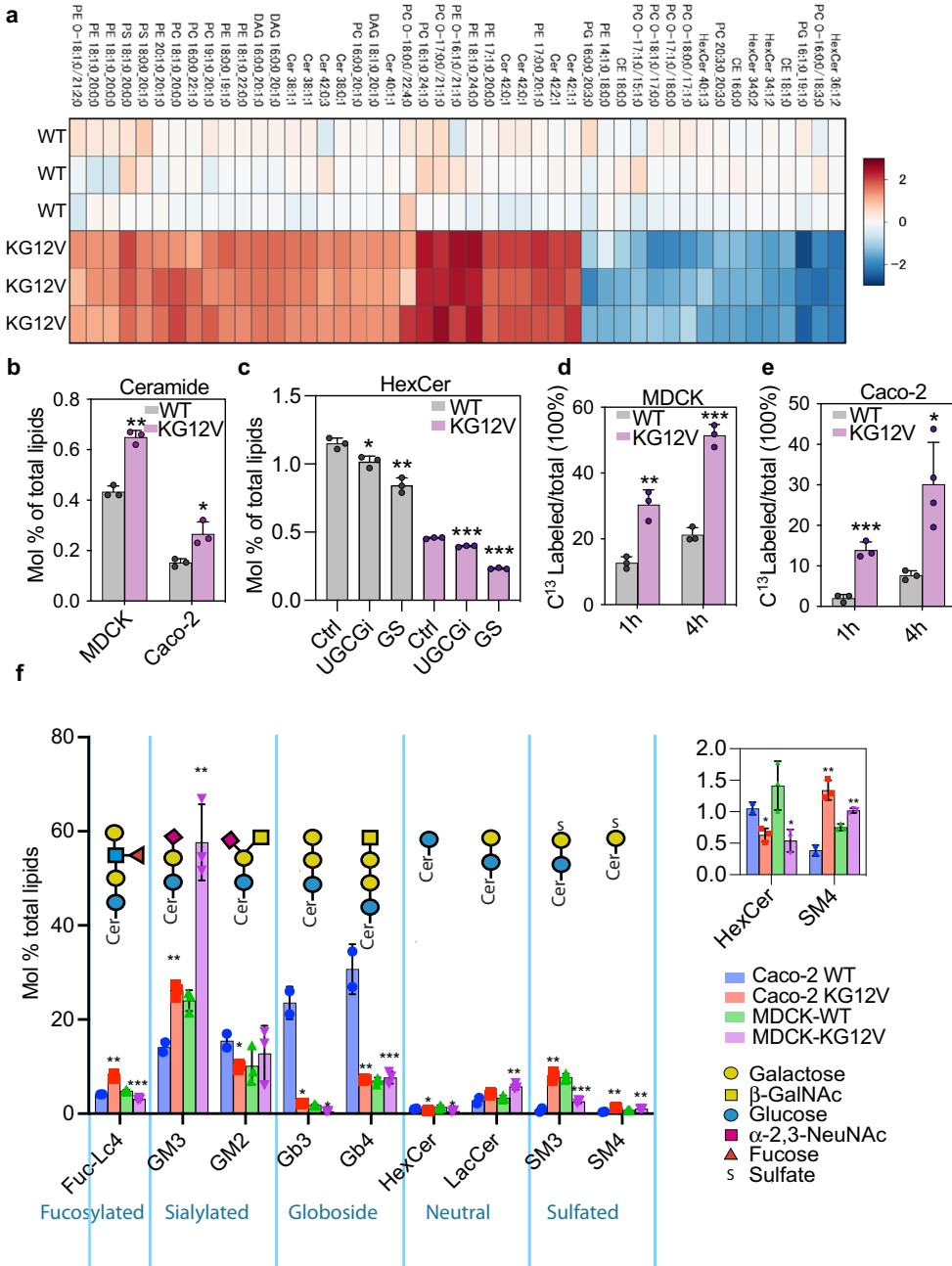

**Fig. 3 | KRASG12V expression reprograms the cellular lipidome.** Lipidomic analysis of parental (WT) versus GFP-KRASG12V-expressing MDCK, or Caco-2 cells: **a** Heatmap of MDCK lipid species that changed most upon KRASG12V expression, shown as log$_2$ fold changes (red = increase, blue = decrease). **b** Molar fractions of ceramide from lipidomic analyses in (**a**). **c** HexCer levels shown as mean mol% in glucose depleted (GS), or DL-PDMP-treated (UGCGi) WT and KRASG12V-expressing MDCK cells. HexCer includes GlcCer and GalCer, which cannot be distinguished by MS unless radiolabeled. **d, e** Fraction of newly synthesized HexCer (=GlcCer) in WT and KRASG12V-expressing MDCK and Caco-2 cells following 1 h and 4 h metabolic labeling with [U-$^{13}$C$_6$]-glucose. **f** Relative abundance of GSLs shown as mol% in WT and KRASG12V-expressing MDCK and Caco-2 cells. Inset shows an enlarged portion of the main graph. Data are means ± SEM. (**a**–**e**) are all mean ± SD; from n = 3 biological replicates; differences were evaluated using two-tailed Student's t-tests (*p < 0.05, **p < 0.01,***p < 0.001).

## KRASG12V PM localization requires specific GSLs

The glycolipidome analysis showed that KRASG12V selectively upregulated GM3 and SM4, and downregulated Gb3. To verify a role for these GSLs in KRAS PM localization we conducted addback experiments under conditions of glucose starvation, DL-PDMP treatment, or UGCG knockout. Addback of GM3 fully recovered KRASG12V PM localization and nanoclustering in DL-PDMP-treated, or glucose-starved MDCK cells, whereas a different class of GSL, Gb3 had no effect (Fig. 4a, b). Addback of GM3, but not Gb3 also restored KRASG12V PM localization in Caco-2 cells deleted for UGCG (Fig. 4c, d). GM3, but not Gb3 addback also recovered PM PtdSer content in DL-

PDMP-treated, or glucose-depleted MDCK cells as evidenced by recovery of PM binding of LactC2 (Fig. 4e). No well-characterized inhibitors of enzymes in the GM3 synthetic pathway beyond UGCG have been described, however, deletion of GM3 synthase (GM3S) in Caco-2 cells significantly reduced KRASG12V PM localization and nanoclustering, which was recovered by addback of GM3, or GM2, a derivative of GM3 (Fig. 4f, g). Deletion of UGT8, which generates Gal-Cer, the first step in SM4 synthesis, resulted in significant reductions of KRASG12V PM targeting and nanoclustering in CaoCo2 cells, which were fully restored by addback of SM4 (Fig. S5a, b). Deletion of LCS significantly reduced PM localization and nanoclustering of

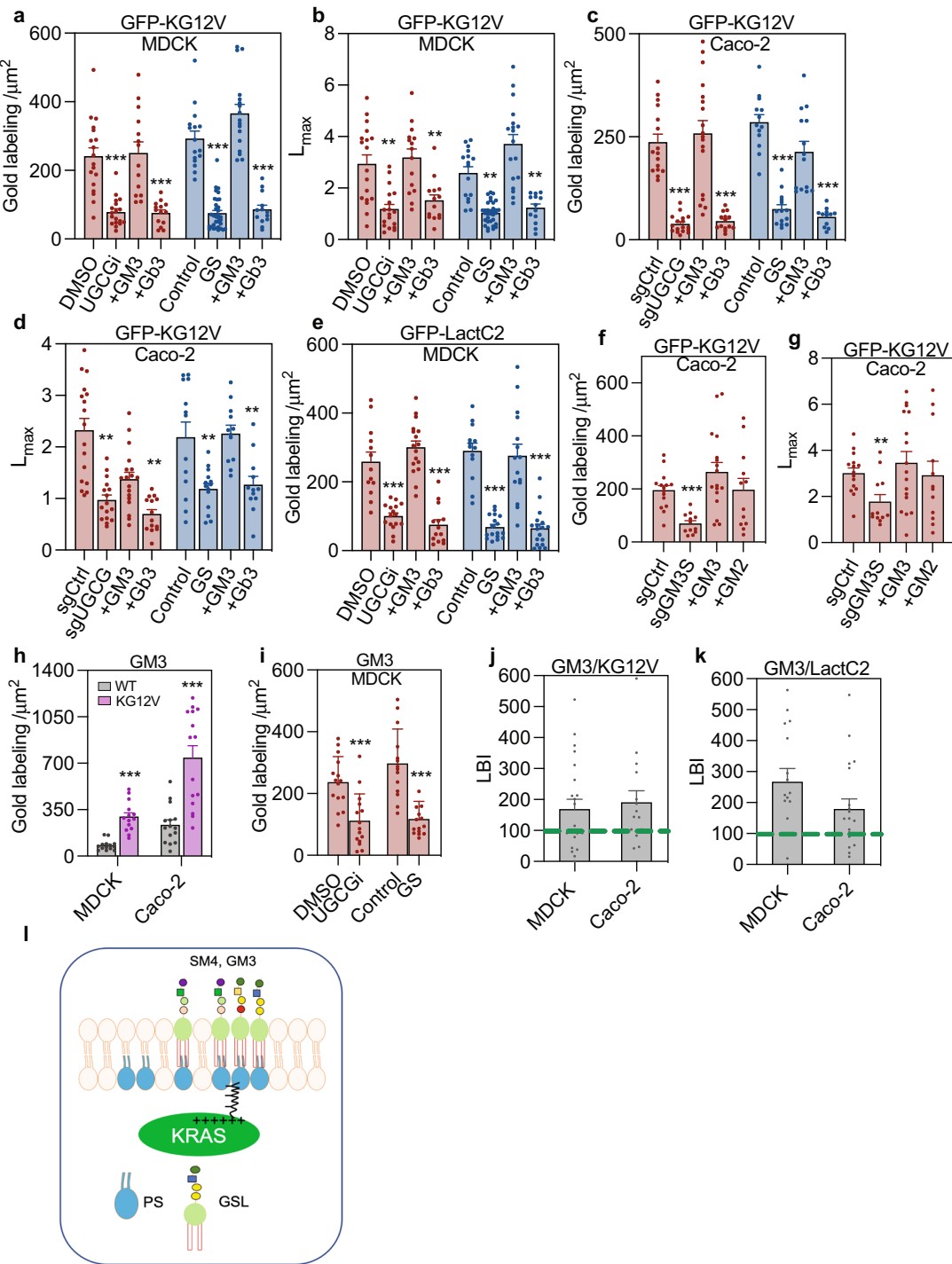

**Fig. 4 | Coupling of outer leaflet GM3 with inner leaflet KRAS and PtdSer is critical for KRAS plasma membrane localization. a, b** EM evaluation of PM sheets of MDCK cells stably expressing KRASG12V (KG12V) and treated with DL-PDMP (UGCGi) ($n = 18$ PM sheets), or cultured in glucose-depleted (GS) ($n = 32$) before incubated with exogenous GM3 ($n = 14$) or Gb3 ($n = 14$) for 1 h. KRASG12V PM binding was quantified as mean gold labeling density and nanoclustering as $L_{max}$. Comparisons are with cognate untreated control cells. **c, d** Identical GSL addback experiments and analyses to (**a**) and (**b**) in KRASG12V-expressing Caco-2 cells. **e** Identical GSL addback experiments to (**a**) in GFP-LactC2-expressing MDCK cells. **f, g** PM localization and nanoclustering of GFP-KRASG12V (KG12V) in Caco-2 cells depleted of GM3 synthase (sgGM3S) ($n = 12$) or expressing control vectors (sgCtrl) ($n = 15$) before incubation with GM3 ($n = 17$) or GM2 ($n = 12$) for 1 h. **h, i** PM sheets from WT ($n = 14$) or KRASG12V-expressing ($n = 15$) MDCK or Caco-2 cells labeled with gold conjugated anti-GM3 antibodies and imaged by EM. PM GM3 content quantified by gold labeling density. **h** PM GM3 levels in WT or KRASG12V-expressing MDCK or Caco-2 cells. **i** PM GM3 levels in DL-PDMP-treated MDCK cells expressing KRASG12V (UGCGi) versus DMSO (control), or glucose-depleted (GS) versus glucose-replete (control). **j, k** PM colocalization of endogenous GM3 and GFP-KRASG12V, or GFP-LactC2 in MDCK ($n = 37$ PM sheets) or Caco-2 cells ($n = 35$ PM sheets) was evaluated by EM and integrated bivariate K-functions (=LBI) after co-labeling with 6-nm-gold-anti-GFP and 2-nm-gold-anti-GM3. LBI > 100 indicates significant colocalization. Differences from control evaluated in bootstrap tests (**$p < 0.01$). **l** A schematic model where outer leaflet GM3 stabilizes KRAS and PtdSer through cross-bilayer coupling. For (**a–i**), data are means ± SEM. Significance of differences between $L_{max}$ and gold labeling density for GSL-addback and control cells were evaluated in bootstrap tests (**$p < 0.01$) or two-tailed Student's t-tests (***$p < 0.001$) respectively. Exact $p$-values and source data are provided as a Source data file.

KRASG12V, which was recovered by GM3, but not SM4 addback (Fig. S5c, d). In MDCK cells cultured with glucose replete medium, addition of GM3, but not SM4 or Gb3, significantly increased PM localization of KRASG12V, suggesting GM3 levels may be limiting for KRAS PM targeting in glucose replete conditions (Fig. S5e). Together these data suggest that GM3 and SM4 are critical for KRAS PM localization and nanoclustering, with GM3 and SM4 being produced by a glycolytic-dependent and glycolytic-independent biosynthetic pathways respectively (Fig. 1a).

Direct quantification of GM3 on the PM by anti-GM3 immunogold labeling and EM further showed that KRASG12V expression significantly elevated the PM levels of GM3 in MDCK and Caco-2 cells, which were reduced to WT levels by glucose depletion, or treatment with DL-PDMP (Fig. 4h, i). Similar results were obtained using indirect histochemistry (Fig. S6a, b), which also allowed validation of the lipid-addback experiments described earlier, since 1 h GlcCer addback, restored surface GM3 in UGCG knockout Caco-2-cells (Fig. S6c, d). Since KRAS PM targeting is dependent on inner leaflet PtdSer, we speculated that outer leaflet GM3 might enhance KRAS PM targeting by forming a complex with inner leaflet PtdSer through cross-bilayer acyl chain coupling[31]. To test this, PM sheets of MDCK or Caco-2 cells expressing GFP-KRASG12V, or GFP-LactC2, were co-labeled with anti-GFP-6 nm-gold and anti-GM3-2 nm-gold and analyzed using bivariate K functions expressed as the summary statistic *LBI*, where *LBI* values greater than 100 (=95% C.I.) indicate significant co-clustering of the 2-nm and 6-nm gold particle distributions. By this assay GM3 extensively colocalized with GFP-KRASG12V and GFP-LactC2 (and hence by inference with PtdSer) in MDCK and Caco-2 cells (Fig. 4j, k). Taken together, these data suggest a model whereby GM3 enhances KRASG12V membrane association and nanoscale organization by forming a complex with inner leaflet PtdSer (Fig. 4l).

### Genetic analysis of GSL synthetic enzymes and KRAS function

Differential expression of GSLs has been reported in different tumor contexts[32], however, the relationship to KRAS oncogenesis has not been explored. We therefore used the GDC-Pan cancer database to evaluate expression profiles of GSL synthases in human cancers with (n = ~730) or without (n = ~11,038) KRAS mutation. *UGCG* and *B4GALT5*, which encodes LCS, were significantly elevated in KRAS mutant cancers (n = ~730) (Figs. 5a, b and S7a), and combined high expression of *UGCG* and *B4GALT5* correlated with shortened 5-year survival (Fig. 5c). These observations suggest positive selection for increased synthesis of GSL precursors in KRAS mutant tumors. Concordant with elevated GM3 levels in KRAS mutant cell lines, high-level expression of *ST3GAL5*, which encodes GM3S, correlated with shortened survival time in KRAS mutant cancers (Fig. 5d). These same tumors exhibited increased expression of *UGT8* and *GAL3ST1* which encode the biosynthetic enzymes for SM4 (Figs. 5a, e and S7b). Elevated expression of *B4GALNT1*, which encodes beta-1,4-N-Acetyl-Galactosaminyltransferase-1 that synthesizes GM2, GD2, GT2, and GA2 from GM3, GD3, GT3, and LacCer, was also correlated with shortened survival in KRAS mutant tumors (Fig. 5f). As a complementary genetic approach we examined whether GSL biosynthesis is also required to support the function of LET-60, the KRAS ortholog in *C.Elegans*. iRNA knockdown of a UGCG ortholog (cgt-3), a UGT8 ortholog (bre-5), or an ortholog of ceramide transport protein (F25H2.6), which transports ceramide to the Golgi for glycosylation, all abrogated LET-60 function as assayed by loss of the multivulva phenotype (Fig. 5g). Concordant with the genetic screen, the UGCG inhibitor DL-PDMP also potently suppressed LET60 function (Fig. 5g).

### GSL synthetic enzymes are required for KRAS oncogenesis

To formally assess the role of GSL metabolism in KRAS oncogenesis, we treated multiple KRAS mutant pancreatic cancer cell lines and a non-transformed pancreatic epithelial cell strain with a UGCG inhibitor. DL-PDMP markedly reduced the viability of KRAS-transformed cancer cell lines: PANC-1, MIA PaCa-2, and MOH, but had no effect on the immortalized, non-transformed pancreatic epithelial cell line: HPNE (Fig. 5h). Next, we used CRISPR-cas9 to delete selected GSL synthases and evaluated colony formation in soft agar. Deletion of LCS, GM3 synthase (ST3GAL5), and B4GALNT1, which generates GM2 and GD2 downstream of GM3 and GD3 respectively (Fig. 5i), all significantly impaired the anchorage-independent growth of PANC1, MOH, and MIA PaCa-2 albeit less robustly in PANC1 cells. In contrast, deletion of ST8SIA1 had no effect on anchorage-independent growth of any cell line. These results strongly implicate GM3 and its derivative GM2, rather than GD2 and GD3 in supporting KRAS oncogenic function (Fig. 5i). Deletion of UGCG and UGT8 also impaired the anchorage-independent growth of PANC-1, MIA PaCa-2, and MOH cells (Fig. 5i). To evaluate KRAS oncogenesis in vivo, we examined the capacity of MiaPaca2 cells deleted for the same GSL synthases to grow as xenografts in nude mice. Deletion of UGCG almost completed blocked xenograft tumor growth (Fig. 6a, b), an effect that was much more robust than the effect on in vitro anchorage-independent growth. Histochemistry of tumor sections also showed that deletion of UGCG significantly reduced levels of phosphorylated MEK (p-MEK) and p-ERK but did not affect levels of ribosomal p-S6 and p-AKT, indicative of selective downregulation of RAF-MEK-ERK signaling cascades (Fig. S8). Similarly, deletion of LCS (B4GALT5), or GM3 synthase (ST3GAL5) markedly reduced the growth of MIA PaCa-2 xenografts, as evident by reductions in tumor weight and size (Fig. 6c–f). By contrast, deletion of GM3 synthase had no effect on the growth of BxPC-3 cell xenografts (Fig. 6g, h). This is an important specificity control since BxPC-3 cells are wild type for KRAS, but exhibit constitutive MAPK activation by virtue of a mutant BRAF allele. Together these observations indicate that GM3 is required to support KRAS oncogenesis, but this requirement is relieved by KRAS-independent activation of MAPK signaling. A selective requirement for GM3 in KRAS oncogenesis is further supported by an absence of synergy between UGCG inhibitors and direct KRAS inhibitors in MIA PaCa-2 cells (Fig. S9). Finally, we used a mouse orthotopic pancreatic ductal adenocarcinoma (PDAC) model where (KRas^LSL.G12D/+;p53^R172H/+;PdxCre) KPC cells stably expressing luciferase were allografted into the pancreas of syngeneic mice and monitored by in vivo luminescent imaging. Concordant with the xenograft experiments, deletion of UGCG in KPC cells significantly reduced tumor growth compared to controls (Fig. 6i, j) and prolonged survival of the orthotopically transplanted mice (Fig. 6k).

## Discussion

We show here that glycolysis controls KRAS PM interactions through defined components of the glycolipidome. Mechanistic experiments directly link GM3 to the maintenance of PM PtdSer content and hence KRAS PM localization and nanoclustering. Enzymes responsible for the biosynthesis of GM3 are essential for activated KRAS signaling and for KRAS oncogenesis in mouse models of pancreatic cancer. Concordantly elevated expression of these same biosynthetic pathways is observed in KRAS mutant cancers and correlate with poor clinical outcomes. Although multiple associations between ganglioside levels and specific tumors have been reported[20,21,33,34], GSL synthase expression patterns and the glycolipidome in the context of KRAS mutational status have not previously been addressed. Together these data indicate that KRAS function is intrinsically linked to GM3 expression, and that one key function of enhanced glycolysis triggered by oncogenic KRAS is elevated GM3 synthesis that maintains, or enhances, KRAS PM localization and oncogenic function in a positive feedforward loop (Fig. 7). We discovered a parallel role for SM4 a GSL classically associated with essential functions in the central nervous system. SM4 expression was consistently increased upon oncogenic KRAS expression, and

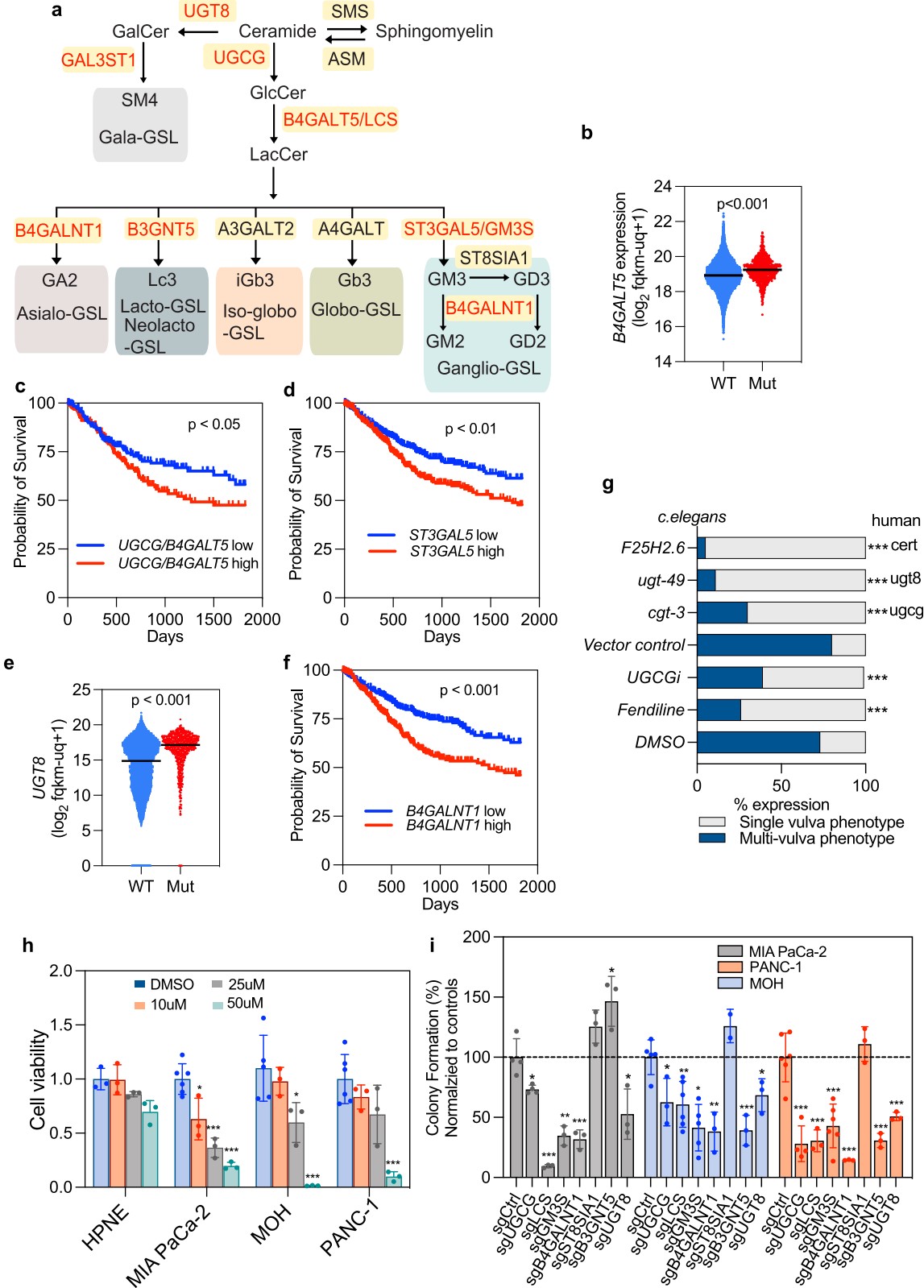

knockdown of the SM4 biosynthetic enzymes abrogated KRAS PM localization and function. Since the key metabolic substrate feeding this pathway is UDP-galactose, itself a derivative of activated glucose, we conclude that a similar positive feedforward loop is operating between KRAS and SM4. Thus, two GSLs, GM3, and SM4, that are required for KRAS PM targeting are elevated as

a result of KRAS activation, operating in parallel with elevated ceramide synthesis, seen as part of a more extensive rewiring of lipid biosynthesis[35–37].

Inhibition of GM3 biosynthesis resulted in a significant reduction of inner leaflet PtdSer content. This loss of PtdSer was not caused by the changes in total PtdSer levels, or flipping of PtdSer to the outer

**Fig. 5 | Requirement for GSL synthetic enzymes human cancer cells and *C. elegans*. a** Major GSL synthetic enzymes with cognate GSL products. Enzymes with increased expression in KRAS-mutant cancers, or whose elevated expression correlates with shortened patient survival, are highlighted in red. **b** Comparative expression of *B4GALT5* in cancer patients with WT (n = 11,038) or mutant KRAS (n = 730) evaluated using Welch's t-test. **c, d** 5-year survival probability of patients with: high (n = 194) versus low (n = 191) *B4GALT5* and *UGCG* expression (**c**), and high (n = 362) versus low (n = 350) *ST3GAL5* expression (**d**). **e** Comparative expression of *UGT8* in cancer patients with WT (n = 11038) or mutant KRAS (n = 730) evaluated using Welch's t-test. **f** 5-year survival probability of patients with high (n = 360) versus low *B4GALNT1* (n = 352) expression. **g** RNAi screen of GSL synthetic enzymes (with closest human ortholog) required to support the multivulva phenotype induced by LET-60 activation in *C. elegans*. Significant differences were evaluated in

two-tailed Student's t-test (***p < 0.001). The numbers of worms analyzed for each condition are provided in the Source data file. **h** MIA PaCa-2, BxPC-3, MOH, Panc-1, HPNE cells treated with increasing concentrations of DL-PDMP (UGCGi) prior to counting. Differences in proliferation rate from control (DMSO-treated) evaluated in two-tailed Student's t-tests (±SD, n = 3 wells, *p < 0.05, **p < 0.01, ***p < 0.001). **i** Anchorage-independent cell growth (colony counts relative to control) ±SD (n = number of wells, each point represents one well) of PDAC cells depleted of *UGCG*, *B4GALT5 (LCS)*, *ST3GAL5 (GM3S)*, *B4GALNT1*, *ST8SIA1*, *B3GNT5*, or *UGT8* using CRISPR-cas9. Differences from each cognate control evaluated in two-tailed Student's t-tests (*p < 0.05, **p < 0.01, ***p < 0.001). For **c, d,** and **f**, the log-rank (Mantel-Cox) test was used to compare curves. Exact p-values and source data are provided as a Source data file.

leaflet. Although GSLs are localized to the outer leaflet of the PM, their trans-bilayer acyl chains potentially could organize lipids on the inner leaflet. In this context we observed strong colocalization of GM3 on the outer leaflet with PtdSer and KRASG12V on the inner leaflet, together with preferential synthesis, in KRAS-transformed cells, of GlcCer species with long acyl chains that were incorporated into GM3. These data recall observations with GPI-anchored proteins (GPI-AP), where long acyl chains that exhibit trans-bilayer interactions with PtdSer were essential for GPI-AP nanoclustering[31], and extend previous work that visualized GM1 and GM3 PM nanoclusters[38–40]. Unresolved questions relate to the role of the sugar head groups of GM3 and whether these directly promote self-assembly into higher order GSL structures, or whether lateral assembly is driven by extracellular lectins, or some combination of both[41].

A second potential mechanism whereby GM3 might regulate PtdSer content on the PM would be via the lipid transport proteins ORP5 and ORP8, which transport PtdSer from the ER to PM with the counter-transport of PI4P from PM to ER. Depleting ORP5, or ORP8, or inhibiting PI4KIIIα that generates PM PI4P, all mislocalize KRAS from the PM and suppress KRAS oncogenesis[10,12,13]. Regulation by GM3 would, however, also require cross-bilayer acyl chain coupling since all components of the transport machinery interact exclusively with the inner leaflet of the PM. The genetic knockdown experiments show that both GM3 and SM4 are independently required to maintain KRAS PM targeting, however, the lack of validated reagents precluded a more detailed analysis of the SM4 mechanism.

The interplay between GSL biosynthesis and oncogenic KRAS function is intriguing. Previous work reported increased influx of glycolysis intermediates into hexosamine biosynthesis and non-oxidative pentose phosphate pathways, as well as increased synthesis of UDP-glucose upon KRASG12D expression[16]. These changes in early glycolytic products, directly linked to GSL biosynthesis, together with reports that the glycolysis inhibitor 2DG significantly reduces cellular HexCer levels in leukemia cells[23], strongly support our contention that KRAS diverts glucose flux into GSL pathways. Enzymes linked to protein and lipid glycosylation are also upregulated by oncogenic KRAS, including B4GALT5 which encodes LacCer synthase[16]. Concordantly, we found increased B4GALT5 transcripts in human KRAS mutant cancers and elevated LacCer levels in cells expressing KRASG12V. Upregulation of LacCer synthase and GOLPH3, a Golgi-localized protein that regulates the distribution and levels of LacCer synthase, has recently been reported in other human cancers[42]. In sum, enhanced glycolipid metabolism is an additional consequence of increased glycolysis triggered by mutant KRAS expression that is in turn essential for KRAS oncogenesis.

A feedforward interplay between growth signals and cell metabolism has previously been demonstrated for N-glycosylation of surface glycoproteins. Branching of N-glycans directly regulates the activities and surface retention of glycoproteins including growth factor receptors[43]. Subsequent activation of surface receptors and PI3K-AKT signaling enhances glycolysis and diverts glucose influx into

hexosamine biosynthetic pathways to generate UDP-N-acetylglucosamine (UDP-GlcNAc), the donor for N-glycosylation, further increasing cell surface GlcNAc-branched glycans[44]. This is analogous to oncogenic KRAS enhancing UDP-glucose production for the synthesis of specific glycolipids, which in turn promote KRAS PM targeting and signaling. This positive feed forward loop between activated KRAS signaling and enhanced glycosphingolipid metabolism extends the known functions of the Warburg effect to high-level regulation of KRAS on the plasma membrane (Fig. 7). Increased GSL synthesis in turn represents a vulnerability of KRAS mutant cells that may be amenable to pharmacological exploitation.

# Methods
## Materials
DL-threo-1-Phenyl-2-decanoylamino-3-morpholino-1-propanol (DL-PDMP) was purchased from Enzo Biochem (# BML-SL210-0010) and dissolved in DMSO. AMG510, a KRASG12C inhibitor, was purchased from Selleck Chemicals (S8830). Brain PtdSer (#840032 C), PIP2 (#840046X), C17:0 Gb3 (#860699), GM3 (#860058), C18 GlcCer (#860547), C18:1 LacCer (#860590), C18:1 GalCer (#860596) and brain SM4 (#131305) were purchased from Avanti Polar Lipids, GM2 from Sigma (G8397). Cell culture medium and glucose-free DMEM medium were purchased from Gibco (Cat#11966025). FBS was purchased from GIBCO. Puromycin was purchased from Thermo Fisher Scientific (BP2956-100). Anti-GFP antibodies used for immunogold labeling were made in house, and anti-GM3 antibodies used for immunogold and immunofluorescent labeling were purchased from Amsbio LLC (GMR6).

## Cell culture and transfection
Mardin-Darby canine kidney (MDCK) cells, Gift of Dr. Robert G. Parton (University of Queensland, Australia); KPC cells, gift of Dr. Jennifer Bailey, McGovern Medical School, Houston; Caco-2, MIA PaCa-2, and PANC-1 cells were purchased from American Type Culture Collection; BxPC3 and MOH were provided by Dr. Craig Logsdon at MD Anderson Cancer, Houston, TX. All cell lines were validated by STR analysis by the vendors or donors. Each cell line used in this study was tested and free of mycoplasma. All cell lines were cultured in 37 °C 5% CO$_2$ humidified incubators. MDCK, PANC-1, KPC, Caco-2 cells were maintained in DMEM medium with 10% FBS. MIA PaCa-2 cells were cultured in DMEM containing 2 mM L-glutamine, 10% FBS, and 2.5% horse serum. BxPC-3 and MOH cells were cultured in RPMI-1640 supplemented with 10% FBS. For glucose starvation, MDCK or Caco-2 cells were cultured in glucose-free DMEM medium containing 10% FBS for 4 h. To generate stable cell lines, $4 \times 10^5$ MDCK or Caco-2 cells were seeded in a 3.5 cm dish on day 1. The next day, cells were transfected with GFP-KRASG12V plasmid DNA with a pEF6 promoter using lipofectamine 2000. On day 3, antibiotic blasticidin at 10 μg/ml was used for selection. The cells were then passaged 3–4 times before being diluted to single cells in a 96-well plate. Cell lines that exhibited optimal and uniform GFP-KRASG12V expression were then picked. Typically, 2–3 rounds of serial

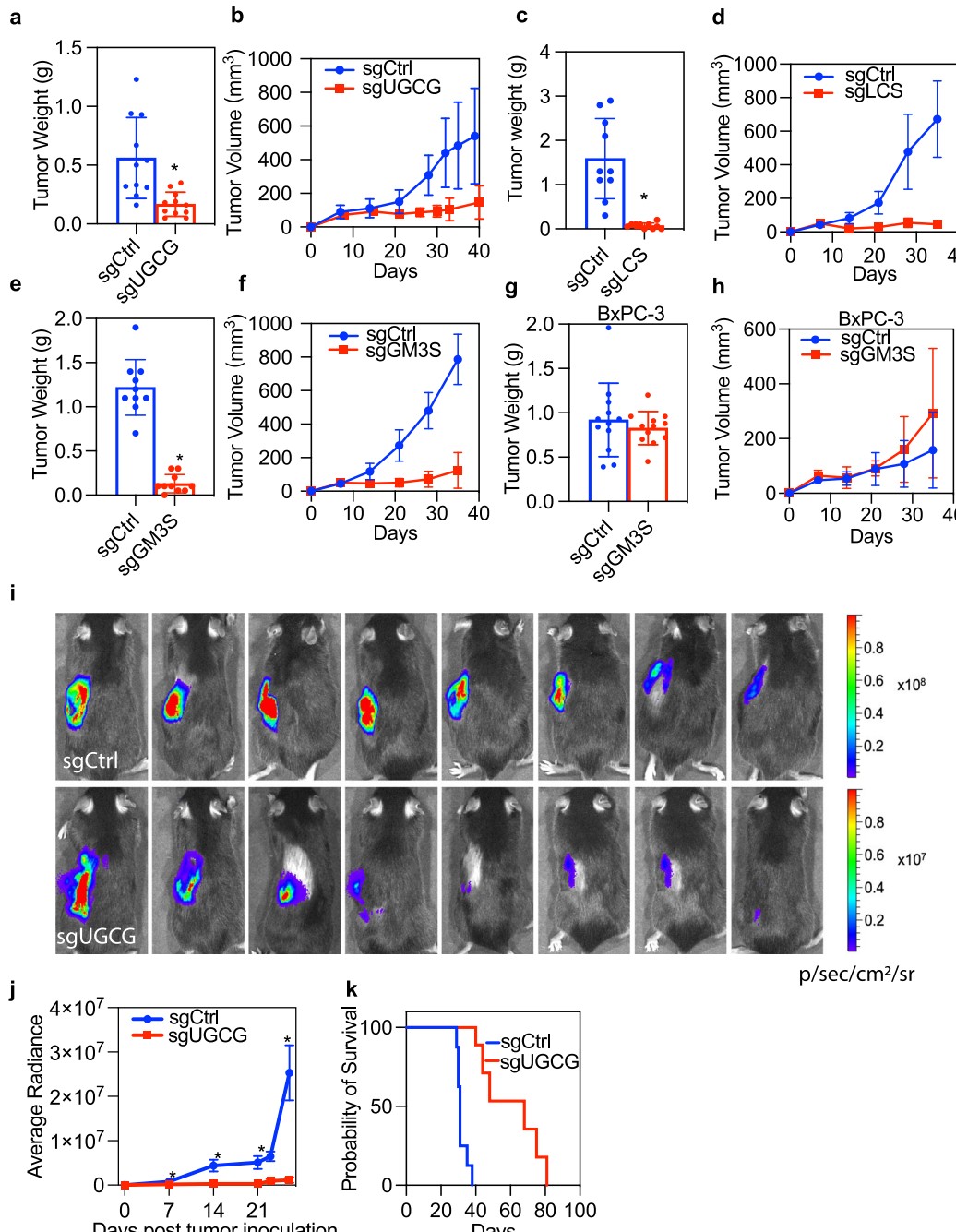

**Fig. 6 | Targeting specific GSL pathways suppresses PDAC tumors in vivo.**
**a**–**f** MIA PaCa-2 cells that stably express control vector (sgCtrl), or sgRNAs targeting UGCG, B4GALT5 (LCS), or ST3GAL5 (GM3S) were injected subcutaneously to opposing flanks of nude mice. Tumor size and volume (±SD, $n = 8$) of the MIA PaCa-2 xenografts were measured: **a**, **b** sgUGCG and control, **c**, **d** sgLCS and control, **e**, **f** sgGM3S and control. **g**, **h** Tumor size and volume (±SD, $n = 8$) of BxPC-3 cells stably expressing sgCtrl or sgGM3S in nude mice. **a**–**h** Differences from each cognate control were evaluated using t-tests (*$p < 0.05$). **i**–**k** Orthotopic implantation of control or UGCG-deleted KPC cells stably expressing luciferase to the pancreas of syngeneic mice. An in vivo imaging system (IVIS) was used to monitor the growth of the orthotopic tumors. **i** Representative images of luminescence signals of a mouse with control or knockout tumors at day 26 post-implantation. **j** Summary of luminescent activity of tumor burden 7–26 days post-implantation. Data are mean ± SEM. **k** Survival curves for mice with control or UGCG knockout tumors. $n = 10$ for each group. Differences between sgCtrl and sgUGCG were evaluated using two-tailed Student's t-tests (*$p < 0.05$).

dilution and confocal imaging were repeated to obtain the ideal monoclonal colonies.

### Cellular ATP measurement
$1 \times 10^6$ MDCK cells were freshly lysed using 100 μl ATP assay buffer from the ATP Assay Kit (Sigma, MAK190) and immediately subject to the following reactions and measurements provided by the vendor.

### Glycolysis assay
$2 \times 10^5$ parental (WT) or KRASG12V-expressing MDCK cells were plated in each well of designated 24-well plates before analyzed by the Seahorse XFe24 analyzer, Agilent. The number of protons exported by cells into the assay medium over a period of 10 min were measured to reflect the basal glycolysis rate. Three wells for each group were used ($n = 3$).

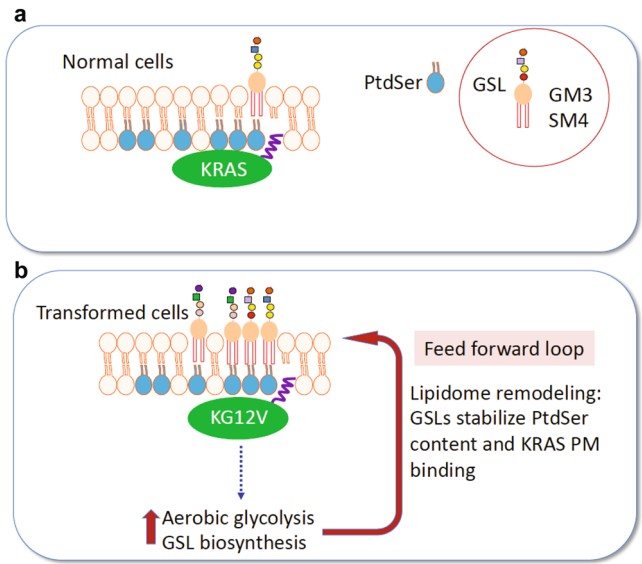

**Fig. 7 | Positive feedback loop between oncogenic KRAS and glycosphingolipid biosynthesis.** A subset of outer leaflet glycosphingolipids, GM3 and SM4, mediate KRAS plasma membrane (PM) interactions. **a** In normal cells these outer leaflet GSLs likely couple with inner leaflet PtdSer through cross-bilayer acyl chain interactions to maintain or stabilize PtdSer content. Inner leaflet PtdSer is in turn essential for KRAS PM binding and nanoclustering. **b** KRAS transformation stimulates increased aerobic glycolysis, the Warburg effect. Oncogenic KRAS diverts this increased glucose influx into GSL biosynthesis and selectively upregulates specific GSLs including GM3 and SM4. This glycolipidome remodeling maintains or promotes KRAS PM targeting. Together these observations identify a feedforward loop between activated KRAS and glycosphingolipid biosynthesis that protects or enhances KRAS signal transmission and which is a potential vulnerability for KRAS oncogenesis.

## Lactate production assay

$2 \times 10^5$ WT or KRASG12V-expressing MDCK cells were seeded in each well of 24-well plate. On the next day, once cell confluence reached 90% percent, we replenished the cell culture medium and incubated cells in the new medium for 6 h. The media were then collected and measured for secreted lactate using the Lactate Assay Kit (Sigma). Three wells for each group were used ($n = 3$).

## Western blotting

Caco-2 cells were washed twice with PBS before incubated on ice with 100 µl lysis buffer containing 1% NP40, 50 mM Tris (pH 7.5), 25 mM NaF, 75 mM NaCl, 5 mM MgCl$_2$, 5 mM EGTA, 0.5 mg/ml aprotinin, 0.1 mM Na$_3$VO$_4$, 3 mg/ml Leupeptin, 1 mM DTT. Samples containing 20 µg protein were separated by SDS-PAGE and blotted onto polyvinylidene difluoride membranes using pre-chilled transfer buffer containing 3.6 mg/ml glycine, 7.25 mg/ml Tris base, 0.46 mg/ml SDS, and 20% methanol. The membranes were then immunoblotted, and signals were detected with enhanced chemiluminescence (Thermo Fisher Scientific) and quantified by a LumiImager (Roche Molecular Biochemicals). The anti-p-ERK, anti-p-MEK, anti-total-ERK, anti-total MEK, anti-β-actin antibodies were purchased from CELL Signaling Technology. All antibodies were used at 1:1000 dilution except β-actin, which was used at 1:4000.

## Transmission electron microscopy

For univariate analyses, basal PM sheets of MDCK and Caco-2 cells were prepared and fixed with 4% PFA and 0.1% glutaraldehyde and labeled with anti-GFP or anti-GM3 antibodies conjugated with 4.5 nm gold, or mCherry antibodies conjugated with 2 nm gold. Images were then taken using JEOL JEM-1400 transmission EM at ×100,000 magnification. ImageJ was used to process the images and to assign x and y coordinates to gold particles in a 1 µm$^2$ area of interest on a PM sheet.

Ripley's K-function was used to quantify the spatial distribution of gold particles under the null hypothesis that all gold points distribute randomly (Eqs. 1 and 2).

$$K(r) = An^{-2} \sum_{i \neq j} w_{ij} \mathbf{1}(||x_i - x_j|| \leq r) \tag{1}$$

$$L(r) - r = \sqrt{\frac{K(r)}{\pi}} - r \tag{2}$$

where $K(r)$ = the univariate K-function for a pattern of $n$ points in a selected PM area of $A$; $r$ = length scale with the range of $1 < r < 240$ nm evaluated at increments of 1 nm; $||\cdot||$ is Euclidean distance; $\mathbf{1}(\cdot)$ is the indicator function with a value of 1 if $||x_i - x_j|| \leq r$, and a value of 0 otherwise; and $w_{ij}^{-1}$ is the fraction of the circumference of a circle with center $x_i$ and radius $||x_i - x_j||$ contained within area $A$, rendering an unbiased edge correction for points at the edge of the study area. $K(r)$ is transformed into $L(r) - r$ which is then normalized on the 99% confidence interval (99% C.I.) estimated via Monte Carlo simulations. Under the null hypothesis of complete spatial randomness $L(r) - r = 0$ for all values of $r$. Values of $L(r) - r$ exceeding the confidence interval indicates significant clustering at that value of $r$. The maximum value of the $L(r) - r$ function, $(L_{max})$ is an unbiased summary parameter that can be used to quantify the extent of clustering[2]. At least 12 PM sheets were imaged, analyzed, and pooled. To evaluate the differences between replicated point patterns a Bootstrap test was employed as described in Eq. (3):

$$D = \sum_{i=1}^{g} \int_{10}^{100} w(r) n_i [K_i(r) - K(r)]^2 dr \tag{3}$$

where $w(r) = r^{-2}$, $K_i(r)$ = weighted mean K-function of the $i$th group of size $n_i$, and $K(r)$ = combined weighted mean K-function of all the groups being compared ($g$). The observed value of D was ranked against 1000 Monte Carlo simulated values of D calculated using a set of residual K-functions derived from each $K_i(r)$[2,45], the rank is an estimate of the probability that the null hypothesis of equivalence can be rejected.

For bivariate analyses, PM sheets of MDCK or Caco-2 cells stably expressing GFP-KRASG12V or GFP-LactC2 were fixed and stained sequentially with anti-GFP 6-nm gold and anti-GM3 2-nm gold conjugated antibodies before imaged and analyzed. Gold particles and colocalizations of GM3/KRASG12V or GM3/LactC2 were calculated with bivariate K-functions (Eqs. 4–7):

$$K_{biv}(r) = (n_b + n_s)^{-1} [n_b K_{sb}(r) + n_s K_{bs}(r)] \tag{4}$$

$$K_{bs}(r) = \frac{A}{n_b n_s} \sum_{i=1}^{n_b} \sum_{j=1}^{n_s} w_{ij} \mathbf{1}(||x_i - x_j|| \leq r) \tag{5}$$

$$K_{sb}(r) = \frac{A}{n_b n_s} \sum_{i=1}^{n_s} \sum_{j=1}^{n_b} w_{ij} \mathbf{1}(||x_i - x_j|| \leq r) \tag{6}$$

$$L_{biv}(r) - r = \sqrt{\frac{K_{biv}(r)}{\pi}} - r \tag{7}$$

where $K_{biv}(r)$ is the weighted mean of two individual bivariate K-functions with $K_{bs}(r)$ which evaluates the spatial distribution of big gold particles relative to each small gold particle and $K_{sb}(r)$ which evaluates the spatial distribution of small gold particles relative to each big gold particle. $K_{biv}(r)$ is transformed to $L_{biv}(r) - r$ and normalized to 95% confidence interval (95% C.I.) estimated under Monte Carlo

simulations. Under the null hypothesis, $L_{biv}(r) - r = 0$ at all values of $r$ indicating no spatial interactions between two gold populations. Positive deviation of the $L_{biv}(r) - r$ exceeding the confidence interval indicate significant colocalizations of two gold patterns at that value of $r$. To quantify the extent of co-clustering, area-under-the-$L_{biv}(r) - r$ curve over a defined scale length ($10 < r < 110$ nm) was calculated using the following equation and was termed as $L_{biv}(r) - r$ integrated or LBI (Eq. 8):

$$LBI = \int_{10}^{110} Std L_{biv}(r) - r.dr. \tag{8}$$

At least 15 PM sheets were imaged, analyzed, and pooled. LBI values ≥100 indicate significant co-clustering of the two gold patterns, and LBI < 100 indicates no co-clustering of the two-point patterns. Although the pooled LBI data are shown as mean ± SEM the LBI parameter is not normally distributed. Therefore, statistical differences between the replicated bivariate point patterns were evaluated in the same Bootstrap test described above (Eq. 3).

## Confocal microscopy
MDCK cells stably co-expressing GFP-KRASG12V and mCherry-CAAX, or GFP-LactC2 and mCherry-CAAX were fixed by 4% PFA prior to imaging in a Nikon Eclipse 80i confocal microscope (Nikon) fitted with a Digital Sight DS-VI1 camera (Nikon) to measure both GFP and mCherry intensity. The Manders coefficient plugin from ImageJ was used for quantification.

## Lipid preparation
Brain PtdSer or $PIP_2$ were dissolved in chloroform, GlcCer, GalCer, GM3, Gb3 in chloroform/methanal (2:1), LacCer in chloroform/methanal/water (5:1:0.1), GM2 in chloroform/methanal/water (2:1:0.1). Vacuum was used to dry the lipids overnight. The next day, lipids were resuspended in complete or glucose-free DMEM medium with 10% FBS before sonicated with Branson 1510 sonicator for 30 min. Medium containing lipids were then kept from light and stored at 37 °C for up to 4 h. MDCK or Caco-2 cells were then cultured with prepared medium containing lipids for 1 h prior to EM analysis.

**MS-shotgun lipidomics.** Mass spectrometry-based lipid analysis was performed by Lipotype GmbH (Dresden, Germany) as described[46]. Lipids were extracted using a two-step chloroform/methanol procedure[47]. Samples were analyzed by direct infusion on a QExactive mass spectrometer (Thermo Scientific) equipped with a TriVersa NanoMate ion source (Advion Biosciences).

## Proliferation assay
PANC-1 ($5 \times 10^3$), MIA PaCa-2 ($2 \times 10^3$), and MOH ($1.5 \times 10^3$) cells[48] were seeded in each well of 96-well plates. The next day, cell culture medium was changed to vehicle (DMSO) or 25 μM DL-PDMP containing medium for 48 h. Cell numbers were then quantified using the Cyquant proliferation assay (Thermo Fisher Scientific) according to the manual.

## Animal experiments
All animal studies were performed under an Institutional Animal Care and Use Committee (IACUC) approved animal protocol, in accordance with the National Institutes of Health Guide for the Care and Use of Laboratory Animals. Female nu/nu (#007850-Outbred athymic nude) and male C57BL/6 (#000664-C57BL/6J inbred) mice were purchased from The Jackson Laboratory (Bar Harbor, ME). Mice with tumors >1.5 cm³ (1.4 cm² diameter), or with an ulcer greater than pinpoint (1-mm) were euthanized. All animals were euthanized by introduction of 100% carbon dioxide to the home cages, followed by cervical dislocation.

## Xenograft experiments
$2.5 \times 10^6$ MIA PaCa-2 cells expressing non-targeting sgRNA (sgCtrl) was injected to the right flanks of nu/nu mice while cells expressing sgUGCG, sgLCS, or sgGM3S were injected to the left flanks, rendering each animal its own control. Tumor volume was measured twice per week by an external caliper and calculated as $V = (length \times width^2)/2$.

## Generation of CRISPR-cas9 cell lines
sgRNAs targeting:
  human UGCG (5′ CCCTGTCTGTCTGCTACGTA 3′),
  B4GALT5 (5′ CAAGTCAAGAGGATAATCTA 3′)
  ST3GAL5 (5′ AACTTCCGGAACCCAAAAGG 3′)
  B4GALNT1 (5′ ACCGGGATGTGTGCGTAGCG 3′)
  UGT8 (5′ GTGGACCCTAATGATATGTG 3′)
  ST8SIA1 (5′ TGCTGCAACAGGGCACGGCG 3′)
  or B3GNT5 (5′ TGCTGCAACAGGGCACGGCG 3′),
  were cloned into pLenti6.3-V5-TOPO vector (K5315-20; Invitrogen) and packed into lentivirus using 293T cells via co-transfection of PMD2G and Pspax2 plasmids. MIA PaCa-2, Panc-1, and MOH cells were then infected with lentivirus encoding sgRNAs for 24 h, followed by selection with 1–4 μg/ml puromycin for 3 passages, to generate stable cell lines. Knockdown efficiency of UGCG was confirmed by western blotting using UGCG antibodies (#128691-AP, Proteintech), and of the rest sgRNAs, was confirmed by real-time PCR using the following primers:
  B4GALT5 (F: CAATCGGTGCTCAGGTTTATG, R: GGTTTCACTG TGGTTCAAGTC)
  ST3GAL5 (F: CCTTCAGTACTCAGAGCCTCAG, R: CTAAGACA ACGGCAATGACACC)
  B4GALNT1 (F: ATTCTTCTTGGATGGGCTTG, R: AGTGATCCTGGG TAACGGTA)
  UGT8 (F: AAGACACCAAGACAAAGCCA, R: GAATTCCCAAGACC CACTCTG)
  ST8SIA1 (F: AATCTCCCTCCTTTGTCAAG, R: GCTCTGTTCCTGT CTTCATA)
  B3GNT5 (F: GCTCTAAACTGCCCTTGAAA, R: GAGATTGCGGAAGA ATGGAA)
  Similarly, KPC cells were transfected with pLenti6.3 encoding sgRNAs targeting mouse UGCG (5′ CATCATGATCTTGTACACAA 3′) and selected by 4 μg/ml puromycin for only 24 h to avoid the integration of cas-9 protein which might elicit immune response in C57BL/6 mice. DNA from single colonies derived from the pooled knockout cells was extracted, amplified by PCR (mUGCG primers: F: CCTTTATGGGATTATATTTACC, R: AGTAGAAGTAGTAGTTGTTG), and sequenced by Sanger sequencing to verify gene knockout.

## KPC orthotopic pancreas implantation model and IVIS imaging
Murine pancreatic adenocarcinoma cells (KPC cells) derived from KRASG12D; Trp53R172H; Pdx1-Cre (KPC) mice of C57BL/6 background were provided by Dr. Jennifer Bailey, McGovern Medical School, Houston, TX. $0.5 \times 10^6$ luciferase-expressing KPC cells that express non-targeting control (sgCtrl) or UGCG targeted (sgUGCG) CRISPR-cas9 vectors, were suspended in 50 μl saline and then injected into the pancreas of C57BL/6 mice by laparotomy. Mice were randomized into two groups: sgCtrl and sgUGCG with 10 mice for each group. 150 mg/kg D-luciferin (# MB102, Syd Labs) was intraperitoneally administered to each mouse and allowed to spread for 2 min followed by 3 min of anesthesia. Mice were then subjected to luminescence imaging using IVIS Lumina XR Imaging System (Caliper Life Sciences). Mice were imaged 1–2 times per week for 3–4 weeks until the first mouse died. A post-implantation survival curve for all mice was subsequently recorded.

## Glycosphingolipidomics by nanoHPLC Chip-Q-TOF MS

$5 \times 10^6$ parental or KRASG12V-expressing MDCK and Caco-2 cells were washed 2 times with PBS before pelletized by high-speed centrifugation. Cells were then subjected to membrane extraction and modified Folch lipid extraction followed by a comprehensive glycosphingolipidome analysis using nanoflow high-performance liquid chromatography chip-quadrupole-time-of-flight mass spectrometry (nanoHPLC Chip-Q-TOF MS) method described in detail by Wong et al.[27].

## Anchorage-independent growth assay

PANC-1 ($5 \times 10^3$), MIA PaCa-2 ($5 \times 10^3$), and MOH ($5 \times 10^3$) parental and sgRNA knockout cells were seeded in soft agar in six-well plates, with a base layer of 1% agar–media mixture, and a top layer of 0.6% agar–cell suspension mix as performed in ref. [49]. After 2–3 wk, colonies were stained with 0.01% crystal violet and imaged. Colony numbers were quantified by CellCounter software.

## *C. elegans* vulva quantification assay

For RNAi experiments, *C. elegans let-60* (n1046) orthologs of human GSL enzymes were researched and identified using the BLAST search and the directory in WormBase, and all RNAi clones from the *C. elegans* RNAi (Ahringer) collection (Source Bioscience) were sequenced and validated. RNAi was introduced by feeding *let-60* worms from L1 larval stage through to the adult stage with *E. coli* HT115 generating double-stranded RNA to target genes of interest. For small molecules, *let-60* (n1046) L1 larvae were cultured in M9 media containing *E. coli* OP50 in the presence of vehicle (DMSO) or 100 mM DL-PDMP, and the extents of the multivulva (Muv) phenotype were scored in adult worms using DIC/Nomarski microscope.

## ¹³C₆-glucose Hex-ceramides profiling by LC-HRMS

To determine the incorporation of glucose carbon into Hex-Ceramides, extracts were prepared and analyzed by high-resolution mass spectrometry (HRMS). Approximately 80% of confluent cells were seeded in 10 cm dishes. Cells were washed with glucose-free medium before incubated in fresh medium containing 11.1 mM $^{13}C_6$-Glucose for 1 and 4 h. Cells were quickly washed with ice-cold PBS to remove extra medium components. Hex-Ceramides were extracted using ice-cold Methanol. Samples were centrifuged at $17,000 \times g$ for 5 min at 4 °C, and organic top layer were transferred to clean tubes, followed by evaporation to dryness under nitrogen. Samples were reconstituted in isopropanol, then 5 μL was injected into a Thermo Vanquish liquid chromatography (LC) system containing an Accucore C30 $2.1 \times 150$ mm column with 2.6 μm particle size. Mobile phase A was 60/40 Acetonitrile/Water and mobile phase B was 90/10 Isopropanol/Acetonitrile. Both A and B contained 10 mM Ammonium formate and 0.1% formic acid. The flow rate was 200 μL/min (at 35 °C), and the gradient conditions were from 40% MPB to 100% MPB in 50 min and hold at 100%B for 10 min. The total run time was 70 min. Data were acquired using a Thermo Orbitrap Fusion Tribrid mass spectrometer under ESI positive ionization mode at a resolution of 240,000. Raw data files were imported into Thermo Trace Finder software for targeted analysis. The fractional abundance of each isotopologue is calculated by the peak area of the corresponding isotopologue normalized by the sum of all isotopologue areas[50].

## Statistics and reproducibility

Prism 9 and Excel Microsoft Excel (version 16.67) were used for two-tailed t-tests and Mantel-Cox test. Statistical details for EM experiments are included in the transmission EM method session. For lipidomics analyses, data are presented as mean ± SD and two-tailed Student's to evaluate the differences between groups. Asterisks are also shown in the respective figures, *$p < 0.05$,

**$p < 0.01$, ***$p < 0.001$. GSL synthetic enzyme mRNA expression and KRAS mutational status in patients and their 5-year survival were analyzed and visualized using data in GDC-PANCAN, by Xena browser, developed by University of California Santa Cruz (https://xenabrowser.net/). The log-rank (Mantel-Cox) test was used to compare the statistical differences between any two survival curves. The in vivo experiments contained 10 mice for each group to generate of 90% power to detect a 20% difference with an estimated variability of 20% using a two-sided t-test at a 5% significance level. All experiments were repeated three times.

## Reporting summary

Further information on research design is available in the Nature Portfolio Reporting Summary linked to this article.

## Code availability

The codes for analyzing EM data are available at a public repository (https://figshare.com/articles/software/Liu_et_al_2022_Nat_Co_codes/21685151).

## Data availability

The lipidomics data generated in this study have been deposited to Figshare (https://figshare.com/articles/dataset/Liu_et_al_lipidomics_xlsx/21678833). The source data underlying all main figures and supplementary figures are provided as a source data file. Specific *p*-values are also included in the source data file. All data are available in the main article, Supplementary Information and source data. Source data are provided with this paper.

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

## Acknowledgements

This work was supported by a Cancer Prevention and Research Institute of Texas (CPRIT) grant (RP200047) to J.F.H.

## Author contributions

J.L., R.V., W.E.K., D.M., W.C., M.W., C.B.L., and Y.Z. performed experiments. J.T.C. analyzed the lipidomics data. J.L. and J.F.H. contributed to the conception of the study, experimental design, data interpretation, and preparation of the manuscript.

## Competing interests

The authors declare no competing interests.
