## [Peer Review File · Nature Communications]

Glycolysis regulates KRAS plasma membrane localization and function through defined glycosphingolipidsREVIEWER COMMENTS

Reviewer #1 (Remarks to the Author):

In their manuscript, Glycolysis regulates KRAS plasma membrane localization and function through defined glycosphingolipids, Lui et al., propose a model through which upregulated glycolytic activity in tumors drives KRAS plasma membrane localization and hence activity, through the specific upregulation of certain glycosphingolipids, specifically GM3 and SM4. They mechanistically dissect this relationship in model cell lines and then validate their findings in CRC- and PDAC-derived cell lines, as well as in a human pancreatic mouse model. As the authors acknowledge in the Discussion previous literature has describe a link between oncogenic KRAS expression and upregulation of early glycolytic events that feed into the GSL biosynthesis pathways implicated in this manuscript. Therefore, I this work proposes an explanation to a very timely and outstanding question in the field. This combination of approaches and novelty makes for a very robust and intriguing study. However, there are some potential inconsistencies, which if addressed could improve the study. I have outlined some suggestions below:

Major

1. There appears to be conflicting data regarding the importance of GalCer for PM localization of KRAS. In Figure 2 GalCer is unable to rescue PM localization and nanoclustering of KRASG12V in glucose starved or DL-PDMP treated MDCK cells. Additionally, GalCer is unable to rescue PM localization and nanoclustering of KRASG12V in UGCG knockout CaCo-2 cells. Finally, GalCer was also unable to raise the PM PtdSer content in glucose-depleted or DP-PMDP-treated MDCK cells. All of this data implicated GSLs downstream of GlcCer as being particularly important for KRASG12V PM localization. In Figure 3, following the analysis of the lipidomics dataset SM4, a derivative of GalCer was upregulated upon KRASG12V expression. The authors should directly discuss this discrepancy. Throughout the rest of the paper much more validation is described for the GM3 GSL than the SM4. Additionally, perturbations of UTG8 rather than GAL3ST1 are applied. Why is this? It is possible that addressing this does not require more experiments, just more clarifications in the text.

2. The authors do a thorough job of documenting the effect of GSL perturbations on KRASG12V membrane localization. Additionally, when they transition to tumor cell line studies, they include HPNE and BxPC3 cells as negative controls. However, it is probable that loss of these GSLs could be affecting other cellular processes independent of RAS. Would it be possible to deplete RAS in some of these models and show that there is no effect of GSL perturbations? Therefore, implying that their role is specifically to target RAS to the membrane. For example, in the proliferation studies in Figure 5 H and I. One would not expect to see synergy between UGCGi and a RAS inhibitor. Or, although RAS KD will decrease colony formation alone, you would not expect to see further reduction in the presence of the sgRNAs that knocked out the essential GSLs in this study.

Minor

1. The inclusion of panel A in Figures 1 and 5 is very useful and was a particularly good idea. It may be worth keeping the color blocks and order for GM3, GA2, LC3, iGb3, and Gb3 consistent between the two figures. Additionally, it would be helpful to include Hex-Cer on the diagram.
2. In Figure 2 is difficult to comprehend for a first-time reader. While it eventually became clear that the first two bars in each group revealed what the perturbation was, it would be easier to comprehend if there was a key for the colors or a label over each set of bars stating the condition being assessed.
3. In the text describing the use of human PDAC cell lines (associated with Figures 5 and 6) the nomenclature is not consistent. They should be referred to using ATCC nomenclature: PANC-1, MIA PaCa-2, BxPC-3. Additionally, CaCo-2 should consistently contain a hyphen, as it does in the figures.
4. I am unaware of the MOH cell line, can a citation be included?

Reviewer #2 (Remarks to the Author):

KRAS is associated with cancers that are highly resistant to treatment. KRAS needs to localize to the plasma membrane, and its clustering at the PM depends on phosphatidylserine (PS). KRAS promotes glycolysis, but whether glycolysis has any role in KRAS signaling is not known. Here, through the use of cell and animal models, the authors demonstrate that glycolysis promotes KRAS signaling through the synthesis of GSLs. Mechanistically, GM3 and SM4 can protein KRAS-PS interaction. Importantly, blocking enzymes in GSL synthesis can abrogate KRAS signaling and cancer growth. Overall, this is an elegant study that provides novel insights to the KRAS signaling and also unveils an additional mechanism by which glycolysis/Warburg promotes cancer growth. The data are also very convincing. Only some minor suggestions.

1. The authors elegantly showed the reduction of inner leaflet PS upon 2DG or DL-PDMP treatment. They argue that this is due to a local effect resulting from a loss of GSLs. However, these treatments could also reduce the synthesis or transport of PS? Also, the addback experiment as shown in Fig. 2E may have increase PS synthesis/delivery. The authors may blot PSS1/2 or ORP5/8 under those conditions. Also the localization of ORP5/8 may be examined. The conclusion of this study is that certain GSLs may impact PS locally, therefore, some global effects on PS synthesis/transport should be ruled out. IF the experiments are hard to do, then discuss this possibility.
2. This is out of curiosity, which may not be directly relevant to this study: what is the effect of glycolysis on PM cholesterol and PS? On cholesterol and PS synthesis?

3. Top of page 10, please refer to figure S8?

Reviewer #3 (Remarks to the Author):

Manuscript: Glycolysis regulates KRAS plasma membrane localization and function through defined glycosphingolipids

In general, the paper is well written and easy to follow, even for a non-expert in GSL biosynthesis. The authors start by using cell lines expressing the KRASG12V mutation, which allows them to study the direct interaction between a RAS mutation and GSL synthesis. After characterizing the connection between RAS and GSL, the authors expand their findings by using cancer cell lines and a mouse model, highlighting the relevance of their results.

Even though most experiments are well conducted, in general, there is a clear lack of detailed methodology. For example, the authors use drugs without ever mentioned drug concentration or test the effect of different drug concentrations (with exception for figure 5).

The authors used two different cell lines (MDCK and Caco-2), which strengthens their findings by showing that the effects are not cell line dependent. However, there is no information about these cell lines. Specifically, they have both cell lines stably expressing GFP-KRASG12V but never provide details on how these cell lines were generated. Additionally, there is no mention of the reason behind using these particular cell lines and not MEFs (the authors cite and compare results to ref.24 where MEFs are used). Finally, there is a lack of consistency on the methodologies used on each cell line. To study GSL biosynthesis, the authors used a drug (DL-PDMP) in MDCK cells, while for the Caco-2 cells they generated a gene deletion using CRISPR. It would be more consistent if at least they show the effect of DL-PDMP in Caco-2 cells.

In summary, this is a well-executed study with strong data to support the claims. Though, an essential aspect that is not addressed regards the specificity of the phenotypes observed after blocking GSL biosynthesis. It is very likely that many other membrane-associated proteins besides KRAS are affected by alterations of the GSL composition. Therefore, are the phenotypic effects reported, i.e. the reduced colony formation in soft agar and diminished Multivulva of the *C. elegans* RAS(gf) mutant, entirely due to reduced membrane recruitment of KRASG12V? If so, then activation of the pathway downstream of RAS, e.g. via activation MEK or MAPK should rescue the effects. This is a central aspect that should be answered in a revised version.

Figure 1 - Suppressing glycolysis mislocalizes KRASG12V from the plasma membrane

- Fig.1A – it is not clear why some components are in black and others in red. Also adding the name of the enzymes would be helpful.

- The authors measured basal glycolysis rate using Seahorse analysis. However, there is no methodology associated with these results. The only information provided is the number of replicates (n=3) which is also not clear to what the authors are referring to (e.g., number of seahorse runs, number of wells analyzed etc).
- In addition to basal glycolysis, it would be interesting to measure lactate levels in WT vs KRASG12V cells. Aerobic glycolysis and Warburg effect (which the authors introduce and discuss) is characterized by an increased glycolytic rate that can only be supported by NAD⁺ replenishment via LDH activity, thus generating lactate.
- As mentioned before, the authors treated the cells with 2DG without providing information on drug concentration or the effect of drug dosage on cell viability. The authors refer that the drug had no observable effect on cell viability but it is unclear how this information is extrapolated from the results. Moreover, in figure 5 the authors clearly show the effect of different drug concentrations in cell viability by measuring colony formation.
- For the localization of KRASG12V on the PM, the authors quantified the number of gold particles and nanoclustering. However, they only present results for treated cells (2DG or DL-PDMP drug treatment), without presenting results for untreated cells though time. It would be interesting to confirm that time does not have an impact on the results.
- The authors start by using 2DG and then moved to glucose starvation (which they use throughout the paper) without an explanation as to why they change methodology. Moreover, glucose starvation could lead to gluconeogenesis where glucose is produced de novo from amino acids and glycerol. This process happens mostly in liver cells, but kidney cells also have this ability. Even though it seems that this process is not affecting the results, the authors should discuss it.
- The authors also do not provide detailed information on how glucose starvation was done.
- The authors apply glucose starvation and measure ATP levels. However, they do not mention the rationale behind measuring ATP levels. Moreover, based on this experiment one could assume that they will use 4 hours glucose starvation for the rest of the experiments. However, this is never mentioned (e.g., in figure 2 cells are exposed to glucose starvation but there is no mention for how long).
- There should be a mention/discussion to the lower levels of ATP after 4 hours.
- Given that 4 hours glucose starvation has no effect on ATP and KRASG12V localization on the PM in MDCK and Caco-2 is measured at 4 hours GS (fig. 1F and G) it is not clear why the authors did a time course for the co-localization of the marker at the membrane (Fig.1H).
- Since the authors did a CRISPR to delete UGCG gene (related with GSL synthesis pathway) it would also be interesting to genetically deplete glycolysis by targeting, for example, HK enzyme.
- Fig.S1 – A and B is swapped in figure legend.
- Fig.S1A shows confocal imaging of MDCK cells expressing GFP-KRASG12V and mCherry-CAAX. However, it is not possible to see the two separate channels as well as the colocalization of the two markers. At 24 hours cells have a clear change in morphology. This is never mentioned or discussed.
- Fig.S1E – WB for p-ERK, ERK, P-MEK and MEK in Caco-2 cells depleted of UGCG. It would be interesting to see the levels also in WT cells. There is no quantification of the WB. In figure legend it says cells were serum starved for 0, 2 or 4h however there is only one WB. There is also no reference

to how many replicates were used. Plus, there is no detail methodology associated with these results.

Fig. 2 – GSLs are required for KRAS plasma membrane localization

- No mention of how long were the cells exposed to DL-PDMP and also GS.
- In this experiment, cells are treated with DL-PDMP or cultured in glucose-depleted medium and incubated with exogenous lipids. However, the authors never add these components to DMSO or control conditions to study their individual effect.

Fig. 3 - KRASG12V expression reprograms the cellular lipidome

- Why is there no Caco-2 data presented in Fig.3A?
- The authors measured HexCer levels (Fig.3C) in starved and UGCGi cells. They should have done the same for ceramide levels.
- In Fig.3C, it is not clear which cell line is being used (MDCK or Caco-2 cells). In the figure legend the authors mention both cell lines but the graph only distinguishes between WT vs KG12V.
- It would be interesting to see the levels of ceramide and HexCer in a RAS depleted cell line. Or in a cell line where RAS does not bind to the membrane.
- The results from ceramide and HexCer do not seem to be in agreement (Fig. 3B and C). The authors suggest that the steady state of HexCer does not necessary reflect its flux and so, they performed an experiment with C13-labeled HexCer where they could see an increased in KG12V. However, it is not clear why was this necessary only for HexCer levels. It would have been interesting to also see this dynamic flux with ceramide. Moreover, figure 3F also does not indicate a dynamic flux of the different components and the results still show that LacCer and different GSLs go up in KG12V. It is really not clear why all the components showed an increase with exception of HexCer.
- The authors should also further explore the differences in results find in Fig.3C, D and E.
- In Fig.3D and 3E the order of magnitude is completely different, with Caco-2 cells having an increase of only 0.2% even though they have the same phenotypes and MDCK cells. The authors acknowledged this fact but present no further discussion or explanation for these differences.

Fig. 4 - Coupling of outer leaflet GM3 with inner leaflet KRAS and PtdSer is critical for KRAS plasma membrane localization

- The authors tested if KRASG12V localization requires specific GSLs. Results showed that GM3 and not Gb3 is important for this localization. However, they did not test all GSL (e.g., LC3, GA2) which opens the question of the role of the other GSL in RAS localization and why focus on GM3 and Gb3 specifically.
- By deleting UGT8 (which generates GalCer) the authors show a reduction of KRASG12V in the PM in Caco-2 cells, which is restored by addback of SM4 (Fig.S5). This led them to concluded that GM3 and SM4 are critical for KRAS PM localization. However, in Fig.2 the authors show that adding GalCer

had no effect in recovering PM localization in glucose starved MDCK or Caco-2 cells. Moreover, the authors also show that addback of GalCer had no impact in raising PM PtdSer (Fig.2E). These results do not seem to be in agreement and the authors ignore this by not presenting any hypothesis or discussing these results.

- To further prove their model where GM3 is important for KRAS localization, the authors directly quantified GM3 at the PM and show that KRASG12V increases the levels of GM3 in MDCK and Caco-2 cells (Fig.4H, I). The authors further explore the role of GM3 by concluding that it forms a complex with inner leaflet PtdSer which regulates RAS localization. However, previously they concluded that both GM3 and SM4 are important for the localization and they also include SM4 on their model (Fig.4L). Which opens the question why only focus on GM3 and not SM4. Also, during discussion, the authors refer to the role of SM4, however, there is not enough data to fully support its involvement, mostly when compared to the evidences provided for GM3.
- In Fig.S6 the authors mention that 1h GlcCer or LacCer but not GalCer addback restored surface GM3 in UGCG Caco-2 cells. However, they only present data for GlcCer.
- Fig.4I – are these data from WT or KRAS cells? in figure legend the authors refer to WT cells, however DMSO or control levels are not identical to the WT values presented in Fig.4H.

Fig.5 - Requirement for GSL synthetic enzymes human cancer cells and *C. elegans*

- What are the expression levels of ST3GAL5 in WT vs RAS cells?
- Fig.5G – *C. elegans* multivulva phenotype. In material and methods, the authors say that worms were exposed to RNAi by feeding through the adult stage. However, multivulva phenotype is determined by VPCs differentiation that occurs during larval development.
- Mention to figure S8 is missing.
- In Fig.5I it is not clear what the different pictures represent. In figure legend the authors say is 26 days post-implantation, so this is time points in hours?

Point by point responses:

Reviewer #1

Major

1. There appears to be conflicting data regarding the importance of GalCer for PM localization of KRAS. In Figure 2 GalCer is unable to rescue PM localization and nanoclustering of KRASG12V in glucose starved or DL-PDMP treated MDCK cells. Additionally, GalCer is unable to rescue PM localization and nanoclustering of KRASG12V in UGGC knockout CaCo-2 cells. Finally, GalCer was also unable to raise the PM PtdSer content in glucose-depleted or DP-PMDP-treated MDCK cells. All of this data implicated GSLs downstream of GlcCer as being particularly important for KRASG12V PM localization. In Figure 3, following the analysis of the lipidomics dataset SM4, a derivative of GalCer was upregulated upon KRASG12V expression. The authors should directly discuss this discrepancy. Throughout the rest of the paper much more validation is described for the GM3 GSL than the SM4. Additionally, perturbations of UGT8 rather than GAL3ST1 are applied. Why is this? It is possible that addressing this does not require more experiments, just more clarifications in the text.

In cells depleted of UGCG, or treated with glucose depleted medium, endogenous SM4 levels are expected to be unaffected as the biosynthetic precursor GalCer is not directly linked to UGCG or UDP-glucose (Figure 1). Cells with sufficient SM4 might not need additional GalCer to feed into the SM4 pool. In contrast, depletion of SM4 by UGT8 deletion renders the cells sensitive to SM4 supplementation; this is shown in the SM4 addback experiment which recovers KRAS plasma membrane targeting in the UGT8 knockout cells (FigS5A, B). Data from human cancer databases (Figure 5) and from glycolipidome analysis (Figure 3) also strongly implicate the relevance of the SM4 in KRAS oncogenesis. The lack of reliable SM4 antibody or other SM4 probes stymied attempts to further examine the role of SM4 using EM spatial mapping as we were able to do with GM3. We chose to deplete UGT8 primarily because it is the functional equivalent of UGCG in the SM4 biosynthetic pathway. We have edited the manuscript to clarify these points as requested by the reviewer.

2. The authors do a thorough job of documenting the effect of GSL perturbations on KRASG12V membrane localization. Additionally, when they transition to tumor cell line studies, they include HPNE and BxPC3 cells as negative controls. However, it is probable that loss of these GSLs could be affecting other cellular processes independent of RAS. Would it be possible to deplete RAS in some of these models and show that there is no effect of GSL perturbations? Therefore, implying that their role is specifically to target RAS to the membrane. For example, in the proliferation studies in Figure 5 H and I. One would not expect to see synergy between UGCGi and a RAS inhibitor. Or, although RAS KD will decrease colony formation alone, you would not expect to see further reduction in the presence of the sgRNAs that knocked out the essential GSLs in this study.

This is an interesting idea; we focused on the use of BxPC-3 and HPNE cells to demonstrate selective specificity for KRAS transformed cells. However, in line with the reviewer's suggestion, we tested whether GSL inhibitors synergize with a KRAS G12C inhibitor. Concordant with the reviewer's speculation, there was no synergy between UGCG inhibitors and the direct RAS inhibitor. These new data results have been included in the supplementary figure S9

Minor

1. The inclusion of panel A in Figures 1 and 5 is very useful and was a particularly good idea. It may be worth keeping the color blocks and order for GM3, GA2, LC3, iGb3, and Gb3 consistent between the two figures. Additionally, it would be helpful to include Hex-Cer on the diagram.

We have modified Figure 1 to keep the color blocks concordant. HexCer includes GlcCer and GalCer, they are grouped together since MS cannot discriminate between the two species, unless the donor sugar is radiolabeled, as in the metabolic labeling experiment. This is now stated in the legend to figure 3.

2. In Figure 2 is difficult to comprehend for a first-time reader. While it eventually became clear that the first two bars in each group revealed what the perturbation was, it would be easier to comprehend if there was a key for the colors or a label over each set of bars stating the condition being assessed.

We have modified Figure 2 as suggested.

3. In the text describing the use of human PDAC cell lines (associated with Figures 5 and 6) the nomenclature is not consistent. They should be referred to using ATCC nomenclature: PANC-1, MIA PaCa-2, BxPC-3. Additionally, CaCo-2 should consistently contain a hyphen, as it does in the figures.

We have made these corrections throughout the text.

4. I am unaware of the MOH cell line, can a citation be included?

MOH cells are a G12R mutant pancreatic cancer cell line gifted by Dr. Craig Logsdon at MD Anderson Cancer Center, Houston, TX. A citation is now included (PMID:30355799).

Reviewer #2:

1. The authors elegantly showed the reduction of inner leaflet PS upon 2DG or DL-PDMP treatment. They argue that this is due to a local effect resulting from a loss of GSLs. However, these treatments could also reduce the synthesis or transport of PS? Also, the addback experiment as shown in Fig. 2E may have increase PS synthesis/delivery. The authors may blot PSS1/2 or ORP5/8 under those conditions. Also, the localization of ORP5/8 may be examined. The conclusion of this study is that certain GSLs may impact PS locally, therefore, some global effects on PS synthesis/transport should be ruled out. If the experiments are hard to do, then discuss this possibility.

Lipidomic analysis showed that DL-PDMP treatment slightly increased the total cellular level of PS, however the increase was not statistically significant. Glucose starvation also increased cellular PS levels; these data are now included in the revised manuscript (Figure S2B) and discussed in the text. Together the data suggest that a reduction in PS synthesis cannot account for the changes in PM PS levels we observe when GSL synthesis is impaired. We have some very preliminary qualitative imaging data showing that DL-PDMP treatment may indeed disrupt the cellular distribution of the PS transporters ORP5 and ORP8. However, we don't have enough quantitative evidence to formally draw this conclusion in the current manuscript. It is an avenue of ongoing investigation. Nevertheless, as requested by the reviewer we now discuss this possible interpretation of the data in a revised discussion.

2. This is out of curiosity, which may not be directly relevant to this study: what is the effect of glycolysis on PM cholesterol and PS? On cholesterol and PS synthesis?

4h of glucose starvation induced a partial loss of PM cholesterol as monitored by the D4H probe, but had no impact on the total cellular level of cholesterol as measured by lipidomics. In contrast, glucose starvation increased the total level of cellular PS as discussed above.

3. Top of page 10, please refer to figure S8?

Reference included.

Reviewer 3

The authors used two different cell lines (MDCK and Caco-2), which strengthens their findings by showing that the effects are not cell line dependent. However, there is no information about these cell lines. Specifically, they have both cell lines stably expressing GFP-KRASG12V but never provide details on how these cell lines were generated. Additionally, there is no mention of the reason behind using these particular cell lines and not MEFs (the authors cite and compare results to ref.24 where MEFs are used).

MDCK cells display distinct features of epithelial cells, including defined junctions and clear apico-basolateral polarity. They grow rapidly and easy to maintain therefore widely used as an epithelial model

cell line. These features render MDCK cells especially suited for high resolution imaging studies. For similar reasons, we chose Caco-2 cells, which originates from human colorectal carcinoma and are readily amenable to CRISPR-cas9 manipulation. How GFP-expressing stable cell lines were generated has now been included in the method.

Finally, there is a lack of consistency on the methodologies used on each cell line. To study GSL biosynthesis, the authors used a drug (DL-PDMP) in MDCK cells, while for the Caco-2 cells they generated a gene deletion using CRISPR. It would be more consistent if at least they show the effect of DL-PDMP in Caco-2 cells.

DL-PDMP has similar effect on KRAS PM targeting in Caco-2 cells. However, to avoid too much repetition we did not include the data in the manuscript. Instead, we used CRISPR-cas9 technique to deplete UGCG as a different means to block GSL synthesis. We reasoned that combining genetic and pharmacological approaches to inhibit UGCG function might be a better way to enhance the rigor of the study.

An essential aspect that is not addresses regards the specificity of the phenotypes observed after blocking GSL biosynthesis. It is very likely that many other membrane-associated proteins besides KRAS are affected by alterations of the GSL composition. Therefore, are the phenotypic effects reported, i.e. the reduced colony formation in soft agar and diminished Multivulva of the *C. elegans* RAS(gf) mutant, entirely due to reduced membrane recruitment of KRASG12V? If so, then activation of the pathway downstream of RAS, e.g. via activation MEK or MAPK should rescue the effects. This is a central aspect that should be answered in a revised version.

The same question was also raised by reviewer #1. To address this, we performed the experiments proposed by reviewer #1 whereby synergy between the direct G12C inhibitor and the UGCG inhibitor was investigated in pancreatic cancer cell line MIA Paca-2 with a G12C mutation. The data revealed that the KRAS inhibitor (AMG510) did not synergize with UGCG inhibitor. We also found that deletion of GM3 synthase did not impact the growth of BxPC3 xenografts. The BxPC3 cell line carries wildtype KRAS alleles but is transformed by mutant BRAF, thus exhibits KRAS independent constitutive MAPK activation. These data strongly suggest that deletion of UGCG or GM3 suppresses cell proliferation through blocking KRAS function, and by extension that constitutive MAPK activation can rescue the effect of UGCG / GM3 synthase inhibition as proposed by the reviewer. That said, these results however do not exclude broader roles for glycosphingolipids on other PM proteins linked to functions beyond KRAS-dependent proliferation for instance cell adhesion, immune recognition.

Fig.1A – it is not clear why some components are in black and others in red. Also adding the name of the enzymes would be helpful.

We agree the coloring was unhelpful have changed the colors of all components to black and added the name of the enzymes in the legend.

The authors measured basal glycolysis rate using Seahorse analysis. However, there is no methodology associated with these results. The only information provided is the number of replicates (n=3) which is also no clear to what the authors are referring to (e.g., number of seahorse runs, number of wells analyzed etc).

We now provide a detailed methodology in the revised paper. The number of replicates refers to number of wells. This is clarified in the figure legend.

In addition to basal glycolysis, it would be interesting to measure lactate levels in WT vs KRASG12V cells. Aerobic glycolysis and Warburg effect (which the authors introduce and discuss) is characterized by an increased glycolytic rate that can only be supported by NAD⁺ replenishment via LDH activity, thus generating lactate.

Increased aerobic glycolysis in KRAS mutant cells is well established and accepted, therefore, we did not extensively quantify aerobic glycolysis using multiple different assays. We did however measure lactate

production in the KRAS transformed cells: data showing a significant increase is now included as Figure S1A.

Even though most experiments are well conducted, in general, there is a clear lack of detailed methodology. For example, the authors use drugs without ever mentioned drug concentration or test the effect of different drug concentrations (with exception for figure 5). (Comment from summary)

As mentioned before, the authors treated the cells with 2DG without providing information on drug concentration or the effect of drug dosage on cell viability. The authors refer that the drug had no observable effect on cell viability, but it is unclear how this information is extrapolated from the results. Moreover, in figure 5 the authors clearly show the effect of different drug concentrations in cell viability by measuring colony formation.

We have modified the text and figure legends to clarify the duration of 2DG treatment and concentration used. Furthermore, in response to the reviewer's later comment about the redundancy of the extended time-course for 2DG and glucose starvation we have amended the figure to just show a 4h time-point for 2DG treatment. The concentration and treatment time is now clearly stated in the legend and methods. In figure 5, we used DL-PDMP not 2DG for the proliferation and colony forming assays.

For the localization of KRASG12V on the PM, the authors quantified the number of gold particles and nanoclustering. However, they only present results for treated cells (2DG or DL-PDMP drug treatment), without presenting results for untreated cells though time. It would be interesting to confirm that time does not have an impact on the results.

We have included the requested control experiments in Fig. S1, where MDCK cells expressing KRASG12V or LactC2 were treated with DMSO for different time durations. No changes in PM localization and nanoclustering of KRASG12V and LactC2 were observed.

The authors start by using 2DG and then moved to glucose starvation (which they use though out the paper) without an explanation as to why they change methodology. Moreover, glucose starvation could lead to gluconeogenesis where glucose is produced de novo from amino acids and glycerol. This process happens mostly in liver cells, but kidney cells also have this ability. Even though it seems that this process is not affecting the results, the authors should discuss it. The authors apply glucose starvation and measure ATP levels. However, they do not mention the rationale behind measuring ATP levels. Moreover, based on this experiment one could assume that they will use 4 hours glucose starvation for the rest of the experiments. However, this is never mentioned (e.g., in figure 2 cells are exposed to glucose starvation but there is no mention for how long). There should be a mention/discussion to the lower levels of ATP after 4 hours.

We have modified the text to address these related points. Many enzymes including kinases are dependent on ATP for activity. Long exposure times to 2DG or glucose starvation reduces ATP levels as cells rely on glycolysis to generate energy. To eliminate the effect of glucose starvation on kinase activities, which might be linked to glycosphingolipid trafficking and synthesis, we chose a time point when cellular ATP was not affected. 2DG and glucose deprivation both significantly mislocalized KRAS from the PM and reduced clustering of KRAS remaining on the PM. We reasoned that glucose starvation is more biologically relevant and highly controlled as addback of glucose to glucose-free medium fully recovered the localization of KRAS to the PM (Fig S1C). The reviewer is correct, we used 4h of glucose starvation throughout the rest of manuscript, since ATP levels were not significantly decreased at this time point. This is clarified in the revised text.

The authors also do not provide detailed information on how glucose starvation was done.

To deprive glucose, the cells were cultured with glucose-free DMEM medium containing 10% Fetal Bovine Serum for varying time durations. We have included this information in the Materials and Methods section.

Given that 4 hours glucose starvation has no effect on ATP and KRASG12V localization on the PM in MDCK and Caco-2 is measured at 4 hours GS (fig. 1F and G) it is not clear why the authors did a time course for the colocalization of the marker at the membrane (Fig.1H).

We agree, on reflection, that showing the extended time course for GS (and 2DG) is redundant after 4h. We have removed these data from the revised paper and just show data at 4h (modified in Fig 1 and Fig S1).

Fig.S1A shows confocal imaging of MDCK cells expressing GFP-KRASG12V and mCherry-CAAX. However, it is not possible to see the two separate channels as well as the colocalization of the two markers. At 24 hours cells have a clear change in morphology. This is never mentioned or discussed.

We have removed the extended time course data from the manuscript in response to the reviewer's valid comment as discussed above. We now show in Fig. S1B, for the control and 4h time point the parallel mCherry-CAAX channel. Quantitation of the pixel by pixel overlap of the two channels (expressed as Manders coefficient) is carried out using an ImageJ plug-in as described in the methods.

Since the authors did a CRISPR to delete UGCG gene (related with GSL synthesis pathway) it would also be interesting to genetically deplete glycolysis by targeting, for example, HK enzyme.

The lack of clinical success of glycolysis inhibitors in treating cancer patients is attributable to high toxicity on normal cells. We therefore focused on identifying novel biosynthetic pathways that are dependent on glycolysis but specific to oncogenic KRAS signaling. We agree with reviewer that it might be interesting to see the impact of genetic manipulations of other glycolytic enzymes on the glycosphingolipid metabolism, but it is beyond the scope of the current manuscript.

Fig.S1 – A and B is swapped in figure legend.

We have corrected this in the legend.

Fig.S1E – WB for p-ERK, ERK, P-MEK and MEK in Caco-2 cells depleted of UGCG. It would be interesting to see the levels also in WT cells. There is no quantification of the WB. In figure legend it says cells were serum starved for 0,2 or 4h however there is only one WB. There is also no reference to how many replicates were used. Plus, there is no detail methodology associated with these results.

We have added the method, corrected the text and quantified the blots.

Fig. 2 –No mention of how long were the cells exposed to DL-PDMP and also GS.

We have now included this information in the legend.

In this experiment, cells are treated with DL-PDMP or cultured in glucose-depleted medium and incubated with exogenous lipids. However, the authors never add these components to DMSO or control conditions to study their individual effect.

We have now measured PM localization of GFP-KRASG12V in MDCK cells cultured with glucose replete media and incubated with different exogenous lipids for 1h. Addition of GM3, but not SM4 or GB3, significantly increased PM localization of KRASG12V, suggesting that GM3 is sufficient to increase KRASG12V PM localization in control conditions. This data is now shown in Fig. S5E and referenced in the text. The data imply that GM3 levels maybe limiting for KRAS PM targeting.

Fig.3 -Why is there no Caco-2 data presented in Fig.3A?

Really just to decompress the figure and avoid too much repetition. The CaCo-2 lipidomic analysis is in the supplementary section. In Fig. S4, we compare the MDCK and Caco-2 lipidomic data and highlight the shared changes upon KRASG12V expression.

The authors measured HexCer levels (Fig.3C) in starved and UGCGi cells. They should have done the same for ceramide levels. The authors should also further explore the differences in results find in Fig.3C, D and E. The results from ceramide and HexCer do not seem to be in agreement (Fig. 3B and C). The authors suggest that the steady state of HexCer does not necessary reflect its flux and so, they performed an experiment with C13-labeled HexCer where they could see an increased in KG12V. However, it is not clear why was this necessary only for HexCer levels. It would have been interesting to also see this dynamic flux with ceramide. Moreover, figure 3F also does not indicate a dynamic flux of the different components and the results still show that LacCer and different GSLs go up in KG12V. It is really not clear why all the components showed an increase with exception of HexCer

UGCGi treatment and glucose starvation did not change the elevated ceramide levels observed in KRASG12V cells, indicating that GSL dependent regulation of PM KRASG12V by UGCG inhibitors and glucose starvation is independent of changes in ceramide synthesis. This data is now shown in Figure S4. Glucose starvation increased cellular ceramide levels in parental cells (WT) but reduced HexCer levels, suggesting that the conversion from ceramide to HexCer was inhibited by glucose starvation. Together these differences likely reflect the increased glycolytic flux into GlcCer synthesis upon oncogenic KRASG12V expression, and the fact that the rate limiting reaction for GSL synthesis is ceramide glycosylation by UGCG.

In Fig.3C, it is not clear which cell line is being used (MDCK or Caco-2 cells). In the figure legend the authors mention both cell lines but the graph only distinguishes between WT vs KG12V.

We have clarified this in the figure legend.

In Fig.3D and 3E the order of magnitude is completely different, with Caco-2 cells having an increase of only 0.2% even though they have the same phenotypes and MDCK cells. The authors acknowledged this fact but present no further discussion or explanation for these differences.

We apologize, Fig. 3E was mislabeled: it should be 20%, not 0.2%, hence the confusion. We have corrected the error, and thank the reviewer for picking this up.

Fig. 4 - The authors tested if KRASG12V localization requires specific GSLs. Results showed that GM3 and not Gb3 is important for this localization. However, they did not test all GSL (e.g., LC3, GA2) which opens the question of the role of the other GSL in RAS localization and why focus on GM3 and Gb3 specifically.

In figure 3F, we showed that GM3 and SM4 increased significantly upon KRASG12V expression in both MDCK and Caco-2 cells. Therefore, we focused on these specific GSLs here. Gene knockdown experiments and GSL addback experiments were then performed to verify the importance of these specific GSLs in KRAS PM targeting. Changes in other GSL expression patterns were variable and so not pursued in this specific context. We have revised the text to make this logic clearer.

By deleting UGT8 (which generates GalCer) the authors show a reduction of KRASG12V in the PM in Caco-2 cells, which is restored by addback of SM4 (Fig.S5). This led them to concluded that GM3 and SM4 are critical for KRAS PM localization. However, in Fig.2 the authors show that adding GalCer had no effect in recovering PM localization in glucose starved MDCK or Caco-2 cells. Moreover, the authors also show that addback of GalCer had no impact in raising PM PtdSer (Fig.2E). These results do not seem be to in agreement and the authors ignore this by not presenting any hypothesis or discussing these results.

This question was also raised by reviewer 1, so please see our response to point 1 for a full discussion. In brief In UGT8 depleted cells, SM4 levels will drop because UGT8 generates GalCer, the precursor of SM4, so rendering cells sensitive to SM4 addback. This is not the case for cells depleted of UGCG, or glucose starved cells, because here GlcCer levels drop, but not GalCer levels, thus SM4 levels will be unaffected by these perturbations.

To further prove their model where GM3 is important for KRAS localization, the authors directly quantified GM3 at the PM and show that KRASG12V increases the levels of GM3 in MDCK and Caco-2 cells (Fig.4H, I). The authors further explore the role of GM3 by concluding that it forms a complex with inner leaflet PtdSer which regulates RAS localization. However, previously they concluded that both GM3 and SM4 are important for the localization and they also include SM4 on their model (Fig.4L). Which opens the question why only focus on GM3 and not SM4. Also, during discussion, the authors refer to the role of SM4, however, there is not enough data to fully support its involvement, mostly when compared to the evidences provided for GM3.

We have presented compelling evidence that both GM3 and SM4 are required to maintain KRAS PM localization. For this reason, both GM3 and SM4 appear on the summary model diagram. That said we concur with the reviewer that the molecular mechanism(s) whereby SM4 exerts this effect is unclear, whereas we can very clearly link GM3 to PM PtdSer content. We focused in this paper on the molecular mechanism of GM3 action because of the availability of a validated antibody to visualize surface expressed GM3, unfortunately similar experiments were not possible for SM4 because we were unable to validate suitable reagents to visualize SM4. In the revised paper, more detailed discussion is provided on the issue of mechanism that both discriminates between what we know about SM4 and GM3, and also offers possible alternate / complimentary mechanisms in response to reviewer 1's request.

In Fig.S6 the authors mention that 1h GlcCer or LacCer but not GalCer addback restored surface GM3 in UGCG Caco-2 cells. However, they only present data for GlcCer.

We have modified the text.

Fig.4I – are these data from WT or KRAS cells? in figure legend the authors refer to WT cells, however DMSO or control levels are not identical to the WT values presented in Fig.4H.

Data are from MDCK-KRASG12V cells. We have modified the figure legend.

Fig.5 - What are the expression levels of ST3GAL5 in WT vs RAS cells?

mRNA levels of St3gal5 slightly increased in Caco-2 cells expressing KRASG12V compared to control cells. Please see the adjacent figure.

Fig.5G The authors say that worms were exposed to RNAi by feeding through the adult stage. However, multivulva phenotype is determined by VPCs differentiation that occurs during larval development.

All RNAi knockdowns were conducted from the L1 larval stage through to the adult stage. We have included this statement in the methods section.

Mention to figure S8 is missing.

We have now added it to the text.

In Fig.6I it is not clear what the different pictures represent. In figure legend the authors say is 26 days post-implantation, so this is time points in hours?

The different pictures are from each individual mouse at day 26. We have clarified the figure legend.

REVIEWERS' COMMENTS

Reviewer #1 (Remarks to the Author):

I feel as though my comments have been adequately addressed, recommend acceptance, and congratulate the authors on a thorough study.

Reviewer #2 (Remarks to the Author):

The authors have added new data and discussion to address my concerns. The manuscript is now acceptable.

Reviewer #3 (Remarks to the Author):

The authors have done a very good job in revising the manuscript and addressed most of the points raised in a satisfactory way. Regarding the key question of specificity for Ras signaling, they included a new experiment showing that simultaneous treatment with GSL and Kras G12C inhibitors does not cause a synergistic effect, and activation of BRAF is insensitive to deletion of GM3 synthetase. This is indirect evidence suggesting that most of the effects of GSL inhibitors are due to reduced KRAS activity. Though other scenarios cannot be excluded, which could be discussed a bit more.